# How Memory in Optimization Algorithms Implicitly Modifies the Loss

**Matias D. Cattaneo**[*]
Princeton University
`cattaneo@princeton.edu`

**Boris Shigida**[*]
Princeton University
`bs1624@princeton.edu`

## Abstract

In modern optimization methods used in deep learning, each update depends on the history of previous iterations, often referred to as *memory*, and this dependence decays fast as the iterates go further into the past. For example, gradient descent with momentum has exponentially decaying memory through exponentially averaged past gradients. We introduce a general technique for identifying a memoryless algorithm that approximates an optimization algorithm with memory. It is obtained by replacing all past iterates in the update by the current one, and then adding a correction term arising from memory (also a function of the current iterate). This correction term can be interpreted as a perturbation of the loss, and the nature of this perturbation can inform how memory implicitly (anti-)regularizes the optimization dynamics. As an application of our theory, we find that Lion does not have the kind of implicit anti-regularization induced by memory that AdamW does, providing a theory-based explanation for Lion's better generalization performance recently documented [13]. Empirical evaluations confirm our theoretical findings.

## 1 Introduction

Many optimization methods used in deep learning are first-order methods with exponentially decaying memory. For example, adding "momentum" to gradient descent (GD) is a well-established practice to make training smoother and convergence faster (e. g. Krizhevsky et al. [39]). Adaptive methods such as Adam [36], RMSProp [61], AdamW [46], and AdaFactor [57], which are commonly used to train large language models [27, 21, 15], all have exponentially decaying memory. Despite the popularity of such optimization methods, there is little theoretical knowledge about the implicit bias memory introduces to them (potentially informing what regions of the loss space the method takes the iterates to, what minima they converge to, how such minima influence the generalization of the trained model, and so on). In this article, we introduce a general framework for identifying such biases.

We study a general class of optimization algorithms described by the following iteration

$$\boldsymbol{\theta}^{(n+1)} = \boldsymbol{\theta}^{(n)} - h\boldsymbol{F}^{(n)}(\boldsymbol{\theta}^{(n)}, \ldots, \boldsymbol{\theta}^{(0)}), \tag{1}$$

where $\boldsymbol{\theta}^{(n)} \in \mathbb{R}^d$ are the (evolving) parameters of the machine learning model, $\boldsymbol{\theta}^{(0)}$ is some initial condition, $h$ is the step size or learning rate, and the functions $\boldsymbol{F}^{(n)}$ map from (some subset of) $(\mathbb{R}^d)^{n+1}$ to $\mathbb{R}^d$ and are allowed to be different at each iteration. The right-hand side in Equation (1) depends on the whole history of previous iterates, which means the algorithm has memory.

---

[*]Authors are listed alphabetically by last name.

39th Conference on Neural Information Processing Systems (NeurIPS 2025).

For many algorithms used in practice, dependence on the history comes in one specific form: by using what we call "*momentum variables*", that is, exponential averages of some functions of the iterate $\boldsymbol{\theta}^{(n)}$ (usually, more specifically, functions of the loss gradient). We present five leading examples to illustrate this point.

*Example* 1.1 (Heavy-ball momentum gradient descent; Polyak [54]). This optimizer can be written in the form (1) with

$$\boldsymbol{F}^{(n)}(\boldsymbol{\theta}^{(n)}, \ldots, \boldsymbol{\theta}^{(0)}) = \boldsymbol{m}_1^{(n+1)},$$

$$\text{where} \quad \boldsymbol{m}_1^{(n+1)} = \sum_{k=0}^{n} \beta^{n-k} \nabla \mathcal{L}(\boldsymbol{\theta}^{(k)}), \tag{2}$$

for some initial condition $\boldsymbol{\theta}^{(0)}$, and where $\beta \in [0, 1)$ is the momentum parameter, $\mathcal{L}$ is the loss function to be optimized, and $\nabla \mathcal{L}$ is its gradient. $\qquad\square$

The optimizer in Example 1.1 is often just referred to as GD with momentum, where the exponential sum $\boldsymbol{m}_1^{(n+1)}$ in Equation (2) is the "momentum variable": it exponentially averages past gradients. The aforementioned optimizer is well-known and often used for training recurrent neural networks and convolutional neural networks, but it underperforms adaptive optimizers when training other architectures such as transformers [68, 45, 2, 40]. The following modification is also commonly used (this formulation is taken from Choi et al. [14] and matches the standard PyTorch implementation).

*Example* 1.2 (Nesterov's accelerated gradient descent; Nesterov [53]). This optimizer can be written in the form (1) with

$$\boldsymbol{F}^{(n)}(\boldsymbol{\theta}^{(n)}, \ldots, \boldsymbol{\theta}^{(0)}) = \boldsymbol{m}_1^{(n+1)} + \boldsymbol{m}_2^{(n+1)},$$

$$\text{where} \quad \boldsymbol{m}_1^{(n+1)} = \beta \sum_{k=0}^{n} \beta^{n-k} \nabla \mathcal{L}(\boldsymbol{\theta}^{(k)}),$$

$$\boldsymbol{m}_2^{(n+1)} = \nabla \mathcal{L}(\boldsymbol{\theta}^{(n)}),$$

for some initial condition $\boldsymbol{\theta}^{(0)}$, and where $\beta \in [0, 1)$ is the momentum parameter, $\mathcal{L}$ is the loss function to be optimized, and $\nabla \mathcal{L}$ is its gradient. $\qquad\square$

The next example presents the most prominent adaptive optimizer, nowadays commonly used for training large language models [27, 21].

*Example* 1.3 (AdamW; Loshchilov and Hutter [46]). The optimizer can be written in the form (1) with

$$\boldsymbol{F}^{(n)}(\boldsymbol{\theta}^{(n)}, \ldots, \boldsymbol{\theta}^{(0)}) = \frac{\boldsymbol{m}_1^{(n+1)}}{\sqrt{\boldsymbol{m}_2^{(n+1)} + \varepsilon}} + \boldsymbol{m}_3^{(n+1)},$$

$$\text{where} \quad \boldsymbol{m}_1^{(n+1)} = \frac{1 - \beta_1}{1 - \beta_1^{n+1}} \sum_{k=0}^{n} \beta_1^{n-k} \nabla \mathcal{L}(\boldsymbol{\theta}^{(k)}),$$

$$\boldsymbol{m}_2^{(n+1)} = \frac{1 - \beta_2}{1 - \beta_2^{n+1}} \sum_{k=0}^{n} \beta_2^{n-k} \left(\nabla \mathcal{L}(\boldsymbol{\theta}^{(k)})\right)^2,$$

$$\boldsymbol{m}_3^{(n+1)} = \lambda \boldsymbol{\theta}^{(n)},$$

for some initial condition $\boldsymbol{\theta}^{(0)}$, and where $0 \leqslant \beta_1, \beta_2 < 1$ are momentum parameters, $\varepsilon > 0$ is a numerical stability parameter, $0 < \lambda < 1$ is a weight decay parameter, and the squares and square roots are taken component-wise. $\qquad\square$

In Example 1.3, $\boldsymbol{m}_1^{(n+1)}$ and $\boldsymbol{m}_2^{(n+1)}$ are also "momentum variables": exponentially averaged gradients and exponentially averaged squared gradient components respectively, with coefficients in front of the sum, such as $(1 - \beta_1)(1 - \beta_1^{n+1})^{-1}$, providing "bias correction" [36]. The variable $\boldsymbol{m}_3^{(n+1)}$ here is a degenerate "momentum variable", with memory decaying infinitely fast.

The following modification incorporates Nesterov's momentum into AdamW. This formulation is taken from Choi et al. [14] (except here $\varepsilon$ is inside the square root in the denominator).

*Example* 1.4 (NAdam with decoupled weight decay; Dozat [22]). The optimizer can be written in the form (1) with

$$\boldsymbol{F}^{(n)}(\boldsymbol{\theta}^{(n)}, \ldots, \boldsymbol{\theta}^{(0)}) = \frac{\beta_1 \boldsymbol{m}_1^{(n+1)} + (1 - \beta_1)\boldsymbol{m}_4^{(n+1)}}{\sqrt{\boldsymbol{m}_2^{(n+1)} + \varepsilon}} + \boldsymbol{m}_3^{(n+1)},$$

$$\text{where} \quad \boldsymbol{m}_1^{(n+1)} = \frac{1 - \beta_1}{1 - \beta_1^{n+1}} \sum_{k=0}^{n} \beta_1^{n-k} \nabla \mathcal{L}(\boldsymbol{\theta}^{(k)}),$$

$$\boldsymbol{m}_2^{(n+1)} = \frac{1 - \beta_2}{1 - \beta_2^{n+1}} \sum_{k=0}^{n} \beta_2^{n-k} \left(\nabla \mathcal{L}(\boldsymbol{\theta}^{(k)})\right)^2,$$

$$\boldsymbol{m}_3^{(n+1)} = \lambda \boldsymbol{\theta}^{(n)},$$

$$\boldsymbol{m}_4^{(n+1)} = \nabla \mathcal{L}(\boldsymbol{\theta}^{(n)}),$$

for some initial condition $\boldsymbol{\theta}^{(0)}$, and where $0 \leqslant \beta_1, \beta_2 < 1$ are momentum parameters, $\varepsilon > 0$ is a numerical stability parameter, $0 < \lambda < 1$ is a weight decay parameter, and the squares and square roots are taken component-wise. □

As a final example, consider a new optimizer called Lion (Evo**L**ved **Si**gn **Mo**me**n**tum), which was recently discovered by an evolutionary search, and then verified to generalize better than AdamW on a variety of tasks [13]. We consider a generalized version of the Lion algorithm.

*Example* 1.5 (Lion-$\mathcal{K}$; Chen et al. [12]). The optimizer can be written in the form of (1) with

$$\boldsymbol{F}^{(n)}(\boldsymbol{\theta}^{(n)}, \ldots, \boldsymbol{\theta}^{(0)}) = -\nabla \mathcal{K}(\boldsymbol{m}_1^{(n+1)} + \boldsymbol{m}_2^{(n+1)}) + \boldsymbol{m}_3^{(n+1)},$$

$$\text{where} \quad \boldsymbol{m}_1^{(n+1)} = -(1 - \rho_2)\frac{\rho_1}{\rho_2} \sum_{k=0}^{n} \rho_2^{n-k} \nabla \mathcal{L}(\boldsymbol{\theta}^{(k)}),$$

$$\boldsymbol{m}_2^{(n+1)} = -\left(1 - \frac{\rho_1}{\rho_2}\right)\nabla \mathcal{L}(\boldsymbol{\theta}^{(n)}),$$

$$\boldsymbol{m}_3^{(n+1)} = \lambda \boldsymbol{\theta}^{(n)}, \tag{3}$$

for some initial condition $\boldsymbol{\theta}^{(0)}$, and where $0 \leqslant \rho_1, \rho_2 < 1$ are Lion's momentum parameters, $\lambda > 0$ is a weight decay parameter, $\mathcal{K} \colon \mathbb{R}^d \to \mathbb{R}$ is some convex function, and $\nabla \mathcal{K}$ is its subgradient. □

We choose the letter $\rho$ rather than $\beta$ for Lion's momentum parameters because they are not precisely parameters controlling the speed of exponential decay in "momentum variables", as explained in Section C. Ordinary Lion corresponds to $\mathcal{K}(\boldsymbol{x}) = \|\boldsymbol{x}\|_1$ and $\nabla \mathcal{K}(\boldsymbol{x}) = \text{sign}(\boldsymbol{x})$ in Example 1.5, where the sign function is understood component-wise. We consider the generalized Lion-$\mathcal{K}$ algorithm because it covers a few known algorithms as special cases: see Table 1 and Section 3.1 in Chen et al. [12]. In fact, it also includes Example 1.1 as a special case by taking $\mathcal{K}(\boldsymbol{x}) = \|\boldsymbol{x}\|^2/2$, $\rho_1 = \rho_2$, and $\lambda = 0$, but we will deal with that important specific example separately for clarity.

It is reasonable to expect that adding exponentially decaying memory to an algorithm in such a way as described above (for example, replacing the gradient with exponentially averaged past gradients) changes the optimization dynamics, thereby affecting the performance of the trained model. The technique we introduce identifies *how* the iteration evolution changes when memory is added. This technique starts with an iteration having memory, and replaces it by a memoryless iteration that approximates the original one, provided a correction term is added. Specifically, we start with algorithm (1), and then construct a corresponding new memoryless iteration:

$$\boldsymbol{\theta}^{(n+1)} = \underbrace{\boldsymbol{\theta}^{(n)} - h\boldsymbol{F}^{(n)}(\boldsymbol{\theta}^{(n)}, \ldots, \boldsymbol{\theta}^{(0)})}_{\text{depends on the whole history } \boldsymbol{\theta}^{(n)}, \ldots, \boldsymbol{\theta}^{(0)}}$$

$$\tilde{\boldsymbol{\theta}}^{(n+1)} = \underbrace{\tilde{\boldsymbol{\theta}}^{(n)} - h\big[\boldsymbol{F}^{(n)}(\tilde{\boldsymbol{\theta}}^{(n)}) + \overbrace{\boldsymbol{M}^{(n)}(\tilde{\boldsymbol{\theta}}^{(n)})}^{\text{correction}}\big]}_{\text{only depends on } \tilde{\boldsymbol{\theta}}^{(n)} \text{ (no memory)}}, \tag{4}$$

where we slightly abuse notation and put

$$\boldsymbol{F}^{(n)}(\tilde{\boldsymbol{\theta}}^{(n)}) \equiv \boldsymbol{F}^{(n)}(\underbrace{\tilde{\boldsymbol{\theta}}^{(n)}, \ldots, \tilde{\boldsymbol{\theta}}^{(n)}}_{n+1 \text{ times}}),$$

and where the function $\boldsymbol{M}^{(n)}(\boldsymbol{\theta})$ captures a correction due to the presence of memory. We then prove an explicit bound on the approximation error $\|\boldsymbol{\theta}^{(n)} - \tilde{\boldsymbol{\theta}}^{(n)}\|$, as a function of the learning rate $h$. Interpreting the correction term can sometimes generate predictions on whether memory helps or hurts generalization of first-order methods with momentum.

Our theory only relies on memory decaying sufficiently fast, not necessarily in the form of momentum variables, and thus covers all the examples listed above and many others, while also allowing for both full-batch and mini-batch training. Section 2 first presents a heuristic discussion of our proposed technique focusing on the simplest possible case for clarity: GD with momentum (Example 1.1). Then, Section 3 presents our main theoretical contribution, which we specialize and apply to all the listed examples in Sections C and D.

Depending on specific optimization algorithm considered, our general result can lead to different practical conclusions. As a substantive application, Section 4 studies AdamW (Example 1.3) and Lion-$\mathcal{K}$ (Example 1.5), and demonstrates that Lion does not suffer from the anti-regularization effect that AdamW's memory has, which predicts better generalization of (full-batch) Lion. Section 5 (with details in Section E) discusses further implications of our main theoretical result: constructing modified equations, and identifying implicit regularization by noise in mini-batch training. Section 6 is devoted to limitations and future directions.

Due to compute constraints, we consider a large-scale empirical testing of this theory out of scope of this paper. However, we provide a preliminary empirical illustration in Section J (with some details, compute resources and licenses in Section K).

## 1.1 Notation

We use standard notations for the $\ell_p$ norm of a vector $\|\boldsymbol{v}\|_p = \left(\sum_i |v_i|^p\right)^{1/p}$; the infinity-norm is defined as $\|\boldsymbol{v}\|_\infty = \max_i |v_i|$; finally, the norm without indices is by default Euclidean: $\|\boldsymbol{v}\| \equiv \|\boldsymbol{v}\|_2$. When we write $\boldsymbol{u}_{n,h} = O(g(h))$, where $g(h)$ is some fixed function of $h$ and $\boldsymbol{u}_{n,h}$ is some sequence of vectors possibly depending on $h$, we mean that there is a constant $C$ not depending on $h$ or $n$ such that $\|\boldsymbol{u}_{n,h}\| \leqslant Cg(h)$. We will contract repeating arguments when convenient, e. g. instead of $\boldsymbol{F}^{(n)}(\boldsymbol{\theta}, \ldots, \boldsymbol{\theta})$ we will write just $\boldsymbol{F}^{(n)}(\boldsymbol{\theta})$. We will use notation $\mathcal{L}(\cdot)$ for the loss and $\nabla\mathcal{L}(\cdot)$ for its gradient, $h$ for the learning rate.

## 2 Building Intuition: Memory Regularizes GD with Momentum

We provide a heuristic explanation of our technique, considering the simplest algorithm with exponentially decaying memory: heavy-ball momentum GD (Example 1.1). As explained above, we would like to remove the dependence of the right-hand side in

$$\boldsymbol{\theta}^{(n+1)} = \boldsymbol{\theta}^{(n)} - h\sum_{k=0}^{n} \beta^{n-k}\nabla\mathcal{L}(\boldsymbol{\theta}^{(k)}) \tag{5}$$

on the "past" iterates $\boldsymbol{\theta}^{(n-1)}, \ldots, \boldsymbol{\theta}^{(0)}$, leaving only the dependence on the "current" iterate $\boldsymbol{\theta}^{(n)}$. Let us represent "past" iterates through the "current" one. First, write

$$\boldsymbol{\theta}^{(n-1)} = \boldsymbol{\theta}^{(n)} + h\sum_{b=0}^{n-1} \beta^b \nabla\mathcal{L}(\boldsymbol{\theta}^{(n-1-b)}) = \boldsymbol{\theta}^{(n)} + h\sum_{b=0}^{n-1} \beta^b \nabla\mathcal{L}(\boldsymbol{\theta}^{(n)}) + O(h^2),$$

where the second equality relies on exponential averaging to replace historical iterates with $\boldsymbol{\theta}^{(n)}$, influencing only higher-order terms. Similarly,

$$\boldsymbol{\theta}^{(n-2)} = \boldsymbol{\theta}^{(n-1)} + h\sum_{b=0}^{n-2} \beta^b \nabla\mathcal{L}(\boldsymbol{\theta}^{(n-2-b)})$$

$$= \boldsymbol{\theta}^{(n-1)} + h \sum_{b=0}^{n-2} \beta^b \nabla \mathcal{L}(\boldsymbol{\theta}^{(n)}) + O(h^2),$$

$$= \boldsymbol{\theta}^{(n)} + h \left\{ \sum_{b=0}^{n-1} \beta^b + \sum_{b=0}^{n-2} \beta^b \right\} \nabla \mathcal{L}(\boldsymbol{\theta}^{(n)}) + O(h^2),$$

where the last equality follows by inserting the expression for $\boldsymbol{\theta}^{(n-1)}$. Continue like this up to

$$\boldsymbol{\theta}^{(n-k)} = \boldsymbol{\theta}^{(n)} + h \sum_{l=1}^{k} \sum_{b=0}^{n-l} \beta^b \nabla \mathcal{L}(\boldsymbol{\theta}^{(n)}) + O(k^2 h^2),$$

where the $k^2$ provides an estimate on the accumulation of error terms of order $h^2$.

We have now represented all the historical iterates through the current one. Combining it with Taylor expansion around $\boldsymbol{\theta}^{(n)}$ in Equation (5), we obtain

$$\boldsymbol{\theta}^{(n+1)} = \boldsymbol{\theta}^{(n)} - h \sum_{k=0}^{n} \beta^k \left\{ \nabla \mathcal{L}(\boldsymbol{\theta}^{(n)}) + h \nabla^2 \mathcal{L}(\boldsymbol{\theta}^{(n)}) \sum_{l=1}^{k} \sum_{b=0}^{n-l} \beta^b \nabla \mathcal{L}(\boldsymbol{\theta}^{(n)}) + O(k^2 h^2) \right\}$$

$$= \boldsymbol{\theta}^{(n)} - h \frac{1 + o_n(1)}{1 - \beta} \nabla \mathcal{L}(\boldsymbol{\theta}^{(n)}) - h^2 \frac{\beta[1 + o_n(1)]}{(1 - \beta)^3} \nabla^2 \mathcal{L}(\boldsymbol{\theta}^{(n)}) \nabla \mathcal{L}(\boldsymbol{\theta}^{(n)}) + O(h^3),$$

where $o_n(1)$ terms go to zero exponentially fast as $n \to \infty$. Now using $\nabla^2 \mathcal{L}(\boldsymbol{\theta}) \nabla \mathcal{L}(\boldsymbol{\theta}) = (1/2) \nabla \|\nabla \mathcal{L}(\boldsymbol{\theta})\|^2$, we obtain that heavy-ball momentum GD is close to ordinary GD (no momentum) with a different step size and different loss, given by

$$\boldsymbol{\theta}^{(n+1)} = \boldsymbol{\theta}^{(n)} - \frac{h}{1 - \beta} \nabla \tilde{\mathcal{L}}(\boldsymbol{\theta}), \quad \text{where} \quad \tilde{\mathcal{L}}(\boldsymbol{\theta}) = \mathcal{L}(\boldsymbol{\theta}) + \frac{h\beta}{2(1 - \beta)^2} \|\nabla \mathcal{L}(\boldsymbol{\theta})\|^2. \tag{6}$$

The term $\frac{h\beta}{2(1-\beta)^2} \|\nabla \mathcal{L}(\boldsymbol{\theta})\|^2$ that is added implicitly to the loss by the momentum can be interpreted as implicit regularization. Since $\beta$ is usually taken to be close to one, the term strongly penalizes the squared norm of the gradient. There is empirical evidence that such penalization improves generalization [6, 59, 26]. In fact, this term (up to coefficients) can be interpreted as a first-order approximation of $\ell_2$ sharpness [26], which suggests that it moves the trajectory towards "flatter" minima; this is often thought to improve generalization [25].

A much more fine-grained analysis of this algorithm is provided in [10] where, in particular, the modified loss is described to arbitrary precision rather than just to first order in $h$.

## 3  General Theory: The Effect of Memory

The general form of an optimization algorithm with memory is given by Equation (1). The only property of memory we use is that it (uniformly in $n$) decays exponentially fast, as made precise by Assumption 3.1 below. Openness and convexity of the domain of optimization $\mathcal{D}$, that is, where all $\{\boldsymbol{\theta}^{(n)}\}$ will be assumed to lie, are innocuous assumptions (e.g., $\mathbb{R}^d$ is open and convex); we impose them to avoid technicalities with differentiation and Taylor expansion.

**Assumption 3.1** (Memory Decay). *Let $\mathcal{D}$ be an open convex domain in $\mathbb{R}^d$. Let $\{\boldsymbol{F}^{(n)}(\boldsymbol{\theta}^{(n)}, \dots, \boldsymbol{\theta}^{(0)})\}_{n=0}^{\infty}$ be a family of functions $\mathcal{D}^{n+1} \to \mathbb{R}^d$, two times continuously differentiable on their respective domains, such that for any $n \in \mathbb{Z}_{\geqslant 0}$, $k_1, k_2 \in \{0, \dots, n\}$, $r, i, j \in \{1, \dots, d\}$,*

$$|F_r^{(n)}| \leqslant \gamma_{-1}, \qquad \left| \frac{\partial F_r^{(n)}}{\partial \theta_i^{(n-k_1)}} \right| \leqslant \gamma_{k_1}, \qquad \left| \frac{\partial^2 F_r^{(n)}}{\partial \theta_i^{(n-k_1)} \partial \theta_j^{(n-k_2)}} \right| \leqslant \gamma_{k_1, k_2},$$

*where $\boldsymbol{F}^{(n)} = (F_1^{(n)}, \dots, F_d^{(n)})^{\mathsf{T}}$, and $\gamma_{-1}$, $\gamma_{k_1}$ and $\gamma_{k_1, k_2}$ are families of positive reals (not depending on $n$) satisfying*

$$\sum_{k_1=1}^{\infty} \gamma_{k_1} k_1^2 + \sum_{k_1, k_2=1}^{\infty} \gamma_{k_1, k_2} k_1 k_2 < \infty. \tag{7}$$

## 3.1 Deriving the Memoryless Approximation

By Taylor expansion with the Lagrange remainder,

$$
F_r^{(n)}(\boldsymbol{\theta}^{(n)}, \dots, \boldsymbol{\theta}^{(0)}) - F_r^{(n)}(\boldsymbol{\theta}^{(n)}, \dots, \boldsymbol{\theta}^{(n)})
$$

$$
= \sum_{k=1}^{n} (\boldsymbol{\theta}^{(n-k)} - \boldsymbol{\theta}^{(n)})^{\mathsf{T}} \frac{\partial F_r^{(n)}}{\partial \boldsymbol{\theta}^{(n-k)}} (\boldsymbol{\theta}^{(n)}, \dots, \boldsymbol{\theta}^{(n)})
$$

$$
+ \frac{1}{2} \sum_{k_1, k_2 = 1}^{n} (\boldsymbol{\theta}^{(n-k_1)} - \boldsymbol{\theta}^{(n)})^{\mathsf{T}} \frac{\partial^2 F_r^{(n)}}{\partial \boldsymbol{\theta}^{(n-k_1)} \partial \boldsymbol{\theta}^{(n-k_2)}} (\boldsymbol{\zeta}) (\boldsymbol{\theta}^{(n-k_2)} - \boldsymbol{\theta}^{(n)})
$$

$$
= \sum_{k=1}^{n} (\boldsymbol{\theta}^{(n-k)} - \boldsymbol{\theta}^{(n)})^{\mathsf{T}} \frac{\partial F_r^{(n)}}{\partial \boldsymbol{\theta}^{(n-k)}} (\boldsymbol{\theta}^{(n)}, \dots, \boldsymbol{\theta}^{(n)}) + O(h^2), \tag{8}
$$

where $\boldsymbol{\zeta}$ is some point on the segment between $(\boldsymbol{\theta}^{(n)}, \dots, \boldsymbol{\theta}^{(0)})$ and $(\boldsymbol{\theta}^{(n)}, \dots, \boldsymbol{\theta}^{(n)})$; in the last step we used Assumption 3.1, $\boldsymbol{\theta}^{(n-k_1)} - \boldsymbol{\theta}^{(n)} = O(k_1 h)$, and $\boldsymbol{\theta}^{(n-k_2)} - \boldsymbol{\theta}^{(n)} = O(k_2 h)$.

Next, write

$$
\boldsymbol{\theta}^{(n-k)} - \boldsymbol{\theta}^{(n)} = \sum_{s=n-k}^{n-1} (\boldsymbol{\theta}^{(s)} - \boldsymbol{\theta}^{(s+1)})
$$

$$
= h \sum_{s=n-k}^{n-1} \boldsymbol{F}^{(s)}(\boldsymbol{\theta}^{(s)}, \dots, \boldsymbol{\theta}^{(0)}) = h \sum_{s=n-k}^{n-1} \boldsymbol{F}^{(s)}(\boldsymbol{\theta}^{(n)}, \dots, \boldsymbol{\theta}^{(n)}) + O(k^2 h^2),
$$

where in the last step we used $\boldsymbol{F}^{(s)}(\boldsymbol{\theta}^{(s)}, \dots, \boldsymbol{\theta}^{(0)}) - \boldsymbol{F}^{(s)}(\boldsymbol{\theta}^{(n)}, \dots, \boldsymbol{\theta}^{(n)}) = O((n-s)h)$, which follows from Taylor expansion and Assumption 3.1. Insert this into Equation (8) and use Assumption 3.1 again to continue:

$$
F_r^{(n)}(\boldsymbol{\theta}^{(n)}, \dots, \boldsymbol{\theta}^{(0)}) - F_r^{(n)}(\boldsymbol{\theta}^{(n)}, \dots, \boldsymbol{\theta}^{(n)})
$$

$$
= h \sum_{k=1}^{n} \frac{\partial F_r^{(n)}}{\partial \boldsymbol{\theta}^{(n-k)}} (\boldsymbol{\theta}^{(n)}, \dots, \boldsymbol{\theta}^{(n)})^{\mathsf{T}} \sum_{s=n-k}^{n-1} \boldsymbol{F}^{(s)}(\boldsymbol{\theta}^{(n)}, \dots, \boldsymbol{\theta}^{(n)}) + O(h^2).
$$

*We conclude that the original numerical iteration can be rewritten in the form* (4)*, where the linear in $h$ correction function is defined as* $\boldsymbol{M}^{(n)} = (M_1^{(n)}, \dots, M_d^{(n)})^{\mathsf{T}}$ *with*

$$
\boxed{ M_r^{(n)}(\boldsymbol{\theta}) := h \sum_{k=1}^{n} \frac{\partial F_r^{(n)}}{\partial \boldsymbol{\theta}^{(n-k)}} (\boldsymbol{\theta})^{\mathsf{T}} \sum_{s=n-k}^{n-1} \boldsymbol{F}^{(s)}(\boldsymbol{\theta}). } \tag{9}
$$

The derivation of the memoryless iteration is now complete. Although not a proof yet, it is the first step towards the approximation bound constituting our main theoretical result.

## 3.2 Approximation Bound

An argument similar to the derivation in Section 3.1 can be made to obtain the following result.

**Theorem 3.2** (Memoryless approximation: 1-step error bound). *Under Assumption 3.1, there exists a discrete memoryless iteration $\{\tilde{\boldsymbol{\theta}}^{(n)}\}_{n=0}^{\infty}$ satisfying* (4) *with initial condition $\tilde{\boldsymbol{\theta}}^{(0)} = \boldsymbol{\theta}^{(0)}$, correction function defined in Equation* (9)*, and a constant $C_1$ not depending on $h$, such that*

$$
\sup_{n \in \mathbb{Z}_{\geqslant 0}} \| \tilde{\boldsymbol{\theta}}^{(n+1)} - \tilde{\boldsymbol{\theta}}^{(n)} + h\boldsymbol{F}^{(n)}(\tilde{\boldsymbol{\theta}}^{(n)}, \dots, \tilde{\boldsymbol{\theta}}^{(0)}) \|_{\infty} \leqslant C_1 h^3.
$$

The proof is available in Section F.

The importance of this one-step approximation result is that it allows to bound the global error between the memoryfull iteration $\boldsymbol{\theta}^{(n)}$ and memoryless iteration $\tilde{\boldsymbol{\theta}}^{(n)}$ on a finite time horizon.

**Corollary 3.3** (Global error bound on a finite "time" horizon). *In the setting of Theorem 3.2, let* $\{\boldsymbol{\theta}^{(n)}\}_{n\in\mathbb{Z}_{\geqslant 0}}$ *be the sequence of vectors generated by the iteration in Equation* (1) *with initial condition* $\boldsymbol{\theta}^{(0)}$. *Let* $T \geqslant 0$ *be a fixed "time" horizon. (The number of iterations considered is not* $T$ *but* $\lfloor T/h \rfloor$.) *Then there exists a constant* $C_2$, *depending on* $T$ *but independent of* $h$, *such that* $\max_{n\in[0:\lfloor T/h\rfloor]}\left\|\boldsymbol{\theta}^{(n)} - \tilde{\boldsymbol{\theta}}^{(n)}\right\|_\infty \leqslant C_2 h^2$.

The proof is in Section G.

## 4 Application: the Effect of Memory on AdamW, Lion and Signum

We first study AdamW with memory by an application of Theorem 3.2 and Corollary 3.3. Neglecting coefficients decaying to zero exponentially fast, we have

$$\boldsymbol{\theta}^{(n+1)} = (1 - \lambda h)\boldsymbol{\theta}^{(n)} - h\bigg(\underbrace{\frac{\nabla\mathcal{L}(\boldsymbol{\theta}^{(n)})}{\sqrt{\left(\nabla\mathcal{L}(\boldsymbol{\theta}^{(n)})\right)^2 + \varepsilon}}}_{\approx\,\mathrm{sign}(\nabla\mathcal{L}(\boldsymbol{\theta}^{(n)}))} + \boldsymbol{M}^{(n)}(\boldsymbol{\theta}^{(n)})\bigg),$$

where $\boldsymbol{M}^{(n)}(\boldsymbol{\theta})$ is given by

$$h\left(\frac{\beta_1(1-\beta_1)^{-1} - \beta_2(1-\beta_2)^{-1}}{\left(\left|\nabla\mathcal{L}(\boldsymbol{\theta})\right|^2 + \varepsilon\right)^{1/2}} + \varepsilon\frac{\beta_2(1-\beta_2)^{-1}}{\left(\left|\nabla\mathcal{L}(\boldsymbol{\theta})\right|^2 + \varepsilon\right)^{3/2}}\right)\left(\nabla\|\nabla\mathcal{L}(\boldsymbol{\theta})\|_{1,\varepsilon} + \lambda\nabla^2\mathcal{L}(\boldsymbol{\theta})\boldsymbol{\theta}\right).$$

Here $\|\cdot\|_{1,\varepsilon}$ is the perturbed one-norm defined as $\|\boldsymbol{v}\|_{1,\varepsilon} := \sum_{i=1}^d \sqrt{v_i^2 + \varepsilon}$. Taking $\varepsilon$ to zero, we can write this in the form of preconditioned gradient descent (with decoupled weight decay):

$$\boldsymbol{\theta}^{(n+1)} = (1 - \lambda h)\boldsymbol{\theta}^{(n)} - h\frac{\nabla\tilde{\mathcal{L}}(\boldsymbol{\theta}^{(n)})}{\left|\left(\nabla\mathcal{L}(\boldsymbol{\theta}^{(n)})\right)\right|},$$

where

$$\tilde{\mathcal{L}}(\boldsymbol{\theta}) = \overbrace{\left[1 + \lambda\left(\tfrac{\beta_2}{1-\beta_2} - \tfrac{\beta_1}{1-\beta_1}\right)h\right]\mathcal{L}(\boldsymbol{\theta})}^{\text{rescaled loss}} - h\left(\tfrac{\beta_2}{1-\beta_2} - \tfrac{\beta_1}{1-\beta_1}\right)\overbrace{\left(\|\nabla\mathcal{L}(\boldsymbol{\theta})\|_1 + \lambda\nabla\mathcal{L}(\boldsymbol{\theta})^\mathsf{T}\boldsymbol{\theta}\right)}^{(*)}$$

is the modified loss. Assuming $\beta_2 > \beta_1$, we see that $(*)$ is implicitly anti-penalized. By Theorem 1.1 in [66], full-batch AdamW converges to a KKT point of the constrained optimization $\min_{\|\boldsymbol{\theta}\|_\infty \leqslant 1/\lambda} \mathcal{L}(\boldsymbol{\theta})$. If $\|\boldsymbol{\theta}\|_\infty \leqslant 1/\lambda$, then the norm $\|\nabla\mathcal{L}(\boldsymbol{\theta})\|_1$ dominates the term $\lambda\nabla\mathcal{L}(\boldsymbol{\theta})^\mathsf{T}\boldsymbol{\theta}$ in absolute value, so the main effect of memory is anti-penalizing the one-norm of the gradient. Thus, *if weight decay is sufficiently small, memory anti-regularizes (large-batch) AdamW*. Incidentally, by Lemma 3.8 in that work, $\boldsymbol{\theta}$ is a KKT point of this optimization problem if and only if the constraint is satisfied and $(*) = 0$. This is a generalization of the observation that the correction term is zero if and only if the point is stationary, true of simpler full-batch algorithms (for Adam with $\lambda = 0$ it follows from the above; for full-batch GD with momentum it is clear from (6)).

Consider now Lion-$\mathcal{K}$ (Example 1.5). Neglecting terms going to zero exponentially fast as $n \to \infty$, the memoryless iteration is

$$\boldsymbol{\theta}^{(n+1)} = (1 - h\lambda)\boldsymbol{\theta}^{(n)} - h\left[-\nabla\mathcal{K}\left(-\nabla\mathcal{L}(\boldsymbol{\theta}^{(n)})\right) + \boldsymbol{M}^{(n)}(\boldsymbol{\theta}^{(n)})\right]$$

$$\text{with}\quad \boldsymbol{M}^{(n)}(\boldsymbol{\theta}) = -h\frac{\rho_1}{1-\rho_2}\nabla^2\mathcal{K}(-\nabla\mathcal{L}(\boldsymbol{\theta}))\nabla^2\mathcal{L}(\boldsymbol{\theta})\left[\nabla\mathcal{K}(-\nabla\mathcal{L}(\boldsymbol{\theta})) - \lambda\boldsymbol{\theta}\right].$$

As mentioned above, ordinary Lion is recovered by setting $\mathcal{K}(\boldsymbol{x}) = \|\boldsymbol{x}\|_1$. This function is not differentiable, so let us replace it with the smooth convex approximation $\|\boldsymbol{x}\|_{1,\varepsilon}$, where $\varepsilon$ is a small positive constant. The results of Section 3 can be applied, and the memoryless iteration is

$$\boldsymbol{\theta}^{(n+1)} = (1 - \lambda h)\boldsymbol{\theta}^{(n)} - h\left[\frac{\nabla\mathcal{L}(\boldsymbol{\theta}^{(n)})}{\left(|\nabla\mathcal{L}(\boldsymbol{\theta}^{(n)})|^2 + \varepsilon\right)^{1/2}} + \boldsymbol{M}^{(n)}(\boldsymbol{\theta}^{(n)})\right]$$

$$\text{with}\quad M_r^{(n)}(\boldsymbol{\theta}) = h\frac{\rho_1}{1-\rho_2}\frac{\varepsilon}{\left(|\nabla_r\mathcal{L}(\boldsymbol{\theta})|^2 + \varepsilon\right)^{3/2}}\nabla_r\left[\|\nabla\mathcal{L}(\boldsymbol{\theta})\|_{1,\varepsilon} + \lambda(\nabla\mathcal{L}(\boldsymbol{\theta})^\mathsf{T}\boldsymbol{\theta} - \mathcal{L}(\boldsymbol{\theta}))\right].$$

This term is small as long as $\varepsilon$ is small. Therefore, better generalization of Lion on a number of tasks [13] may be partially attributed to the fact that memory does *not* anti-regularize Lion. In addition, notice that the correction terms are exactly the same for Adam with $\beta_1 = \beta_2 =: \beta$ and Lion with $\rho_1 = \rho_2 = \beta$. Since Lion with $\rho_1 = \rho_2$ is Signum [8], we provide a novel perspective on the similarity between Adam with $\beta_1 \approx \beta_2$ and Signum, a point discussed and verified empirically in [69].

## 5 Further Implications

### 5.1 Modified Equations

We have taken a very general algorithm (1) and converted it (under Assumption 3.1) to a memoryless iteration (4) with $O(h^2)$ uniform error bound on a finite time horizon (Corollary 3.3). Since this iteration has no memory, standard methods can be used to derive an ordinary differential equation (ODE) in the form $\dot{\boldsymbol{\theta}} = \boldsymbol{G}_h(\boldsymbol{\theta}) =: \boldsymbol{G}_1(\boldsymbol{\theta}) + h\boldsymbol{G}_2(h)$ whose continuous solution approximates this iteration and hence the initial algorithm (with the same approximation guarantee). Similarly to Section 2, the derivation is heuristic (although the approximation bound is easily made rigorous) and proceeds as follows. Relating the iteration number $n$ of a discrete iteration and the time point $t = nh$ on a continuous trajectory, we would like the continuous trajectory to satisfy the same one-step relation as the discrete iteration, up to $O(h^3)$:

$$\boldsymbol{\theta}((n+1)h) = \boldsymbol{\theta}(nh) - h\big[\boldsymbol{F}^{(n)}(\boldsymbol{\theta}(nh), \dots, \boldsymbol{\theta}(nh)) + \boldsymbol{M}^{(n)}(\boldsymbol{\theta}(nh))\big] + O(h^3).$$

In fact, we will ensure it is true for $nh$ replaced by any $t$:

$$\boldsymbol{\theta}(t+h) = \boldsymbol{\theta}(t) - h\big[\boldsymbol{F}^{(n)}(\boldsymbol{\theta}(t), \dots, \boldsymbol{\theta}(t)) + \boldsymbol{M}^{(n)}(\boldsymbol{\theta}(t))\big] + O(h^3). \tag{10}$$

But, using a Taylor expansion, and recalling that we are finding the trajectory satisfying $\dot{\boldsymbol{\theta}}(t) = \boldsymbol{G}_h(\boldsymbol{\theta}(t))$, hence $\ddot{\boldsymbol{\theta}}(t) = \nabla\boldsymbol{G}_h(\boldsymbol{\theta}(t))\dot{\boldsymbol{\theta}}(t)$, we have

$$
\begin{aligned}
\boldsymbol{\theta}(t+h) &= \boldsymbol{\theta}(t) + h\dot{\boldsymbol{\theta}}(t) + \frac{h^2}{2}\ddot{\boldsymbol{\theta}}(t) + O(h^3) \\
&= \boldsymbol{\theta}(t) + h\big\{\boldsymbol{G}_1(\boldsymbol{\theta}(t)) + h\boldsymbol{G}_2(\boldsymbol{\theta}(t))\big\} \\
&\quad + \frac{h^2}{2}\big\{\nabla\boldsymbol{G}_1(\boldsymbol{\theta}(t))\boldsymbol{G}_1(\boldsymbol{\theta}(t)) + O(h)\big\} + O(h^3) \\
&= \boldsymbol{\theta}(t) + h\boldsymbol{G}_1(\boldsymbol{\theta}(t))h^2\Big\{\boldsymbol{G}_2(\boldsymbol{\theta}(t)) + \frac{\nabla\boldsymbol{G}_1(\boldsymbol{\theta}(t))\boldsymbol{G}_1(\boldsymbol{\theta}(t))}{2}\Big\} + O(h^3).
\end{aligned}
$$

In order to match (10), we need

$$
\begin{aligned}
\boldsymbol{G}_1(\boldsymbol{\theta}) &= -\boldsymbol{F}^{(n)}(\boldsymbol{\theta}, \dots, \boldsymbol{\theta}), \\
\boldsymbol{G}_2(\boldsymbol{\theta}) &= -\Big(\boldsymbol{M}^{(n)}(\boldsymbol{\theta})/h + \frac{\nabla\boldsymbol{G}_1(\boldsymbol{\theta})\boldsymbol{G}_1(\boldsymbol{\theta})}{2}\Big).
\end{aligned}
$$

So, apart from the correction term coming from memory, the ODE $\dot{\boldsymbol{\theta}} = \boldsymbol{G}_1(\boldsymbol{\theta}) + h\boldsymbol{G}_2(\boldsymbol{\theta})$ derived has another term

$$h^2\frac{\nabla\boldsymbol{G}_1(\boldsymbol{\theta})\boldsymbol{G}_1(\boldsymbol{\theta})}{2}$$

arising from the fact that the algorithm is discrete.

For the example of full-batch heavy-ball momentum GD as in Section 2, where $\boldsymbol{G}_1(\boldsymbol{\theta}) = -(1-\beta)^{-1}\nabla\mathcal{L}(\boldsymbol{\theta})$ (ignoring coefficients going to zero exponentially fast in $n$), this additional term is equal to $h^2(1-\beta)^{-2}\nabla\|\nabla\mathcal{L}(\boldsymbol{\theta})\|^2/4$, providing additional implicit regularization. We recover the ODE derived by Kovachki and Stuart [37], Ghosh et al. [26]:

$$\dot{\boldsymbol{\theta}} = -\frac{\nabla\mathcal{L}(\boldsymbol{\theta})}{1-\beta} - h\frac{1+\beta}{(1-\beta)^3}\frac{\nabla\|\nabla\mathcal{L}(\boldsymbol{\theta})\|^2}{4}.$$

## 5.2 Mini-Batch Training

In specific cases, it is possible to identify the additional implicit regularization that is introduced to the algorithm by noise, if mini-batch training is used as opposed to full-batch. Assume that the form of $\boldsymbol{F}^{(n)}(\boldsymbol{\theta}^{(n)}, \ldots, \boldsymbol{\theta}^{(0)})$ is given by

$$\boldsymbol{F}^{(n)}(\boldsymbol{\theta}^{(n)}, \ldots, \boldsymbol{\theta}^{(0)}) = \sum_{k=0}^{n} \beta^k \boldsymbol{g}^{(\pi(n-k))}(\boldsymbol{\theta}^{(n-k)}),$$

where the $\{\boldsymbol{g}^{(k)}(\cdot)\}_{k=0}^{n}$ functions are uniformly bounded along with two derivatives, and $\pi$ is a random permutation of $(0, \ldots, n)$ (chosen among all such permutations with equal probability). The interpretation is that $n$ is a large number of mini-batches in one epoch, and mini-batches are sampled randomly without replacement.

The correction term introduced by memory (9) is

$$M_r^{(n)}(\boldsymbol{\theta}) = h\beta \sum_{k=0}^{n-1} \beta^k \nabla g_r^{(\pi(n-1-k))}(\boldsymbol{\theta})^\mathsf{T} \sum_{l=1}^{k+1} \sum_{b=0}^{n-l} \beta^b \boldsymbol{g}^{(\pi(n-l-b))}(\boldsymbol{\theta}).$$

We can take the average $\mathbb{E}$ over all permutations $\pi$ (re-orderings of mini-batches). Deferring some details to Section E, the result is that for large $n$

$$\mathbb{E}[M_r^{(n)}(\boldsymbol{\theta})]/h \approx \frac{\beta}{(1-\beta)^3} \nabla g_r(\boldsymbol{\theta})^\mathsf{T} \boldsymbol{g}(\boldsymbol{\theta})$$

$$+ \frac{\beta}{(1-\beta)^2(1+\beta)} \mathbb{E}[(\nabla g_r^{(\pi(1))}(\boldsymbol{\theta}) - \nabla g_r(\boldsymbol{\theta}))^\mathsf{T}(\boldsymbol{g}^{(\pi(1))}(\boldsymbol{\theta}) - \boldsymbol{g}(\boldsymbol{\theta}))].$$

The second term can be interpreted as implicit regularization by noise. For clarity, $\pi(1)$ is a uniformly distributed random variable over $\{0, \ldots, n\}$, so this expectation is an average over mini-batch indices.

For example, take $\boldsymbol{g}^{(k)}(\boldsymbol{\theta}) = \nabla \mathcal{L}^{(k)}(\boldsymbol{\theta})$ the $k$th minibatch loss. Then we obtained that "on average" mini-batch GD with momentum is given by the iteration like (6), except the modified loss has an additional regularization term:

$$\tilde{\mathcal{L}}(\boldsymbol{\theta}) = \mathcal{L}(\boldsymbol{\theta}) + \frac{h\beta}{2(1-\beta)^2} \|\nabla \mathcal{L}(\boldsymbol{\theta})\|^2 + \underbrace{\frac{h\beta}{2(1-\beta)(1+\beta)} \mathbb{E}\|\nabla \mathcal{L}^{(\pi(1))}(\boldsymbol{\theta}) - \nabla \mathcal{L}(\boldsymbol{\theta})\|^2}_{\text{regularization by mini-batch noise}}.$$

# 6 Limitations and Future Directions

The approximation bounds in Section 3.2 are in terms of the learning rate $h$, which means that $h$ has to be sufficiently small for (our approximations and hence) the optimization trajectories to be close. This is a standard limitation in the literature. Fortunately, practically relevant learning rates are often small enough (especially mid-training, if a learning rate decay schedule is used). Additionally, there is a non-negligible effect of mini-batch noise on the picture we are describing in Section J; in particular, Lion does not necessarily outperform Adam if batches are small [13]. We are able to characterize this effect using similar techniques, but this is out of scope of this article and is a work in progress.

Taking a broader view, one may question the effect of (explicit or implicit) regularization on training progress and outcomes in deep learning, which is an intricate question not easily amenable to theoretical analysis [67, 1, 34]. Between 2023 and 2025, the mainstream paradigm in deep learning (especially large language models) is best described as scaling-centric rather than generalization-centric [65] which decreased interest in implicit regularization. The main purpose of this work is to introduce a general framework for identifying correction terms, whether they are treated as implicit regularizers or not. In the future, it is likely possible to build on it to study implications for the training dynamics, including characterizing the properties of the loss landscape around the optimizer's trajectory. Additionally, it is reasonable to anticipate that because of limited high-quality data, overfitting is about to become a widely important issue again, if not already [65, Section 7], [62, 35].

Finally, we discuss some of the most popular optimizers in recent years, but other important algorithms like Shampoo [30, 58] or its versions are also amenable to this analysis, and the approximation results in Section 3.2 hold for them (assuming a typical choice of momentum schemes). However, interpreting the higher-order corrections is not trivial, and we leave that as additional future work.

## Acknowledgments

We thank Boris Hanin for his comments. Cattaneo gratefully acknowledges financial support from the National Science Foundation through DMS-2210561 and SES-2241575. We acknowledge the Princeton Research Computing resources, coordinated by the Princeton Institute for Computational Science and Engineering (PICSciE) and the Office of Information Technology's Research Computing.

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

# A    Related Literature

Approximating a memoryful iteration with a memoryless one is closely connected with the method of modified equations (sometimes called *backward error analysis*), where a discrete algorithm like (1) is approximated by a continuous solution of an ordinary differential equation or stochastic differential equation. Typically, this method can only be applied to an algorithm with no memory, in a possibly enlarged phase space as opposed to $\mathbb{R}^d$; for example, heavy-ball momentum GD has no memory when viewed as a discrete iteration $(\boldsymbol{\theta}^{(n)}, \boldsymbol{m}^{(n)})$ in $\mathbb{R}^{2d}$, where $\boldsymbol{m}^{(n)}$ is the "momentum variable". The general technique introduced in this paper can be used to derive a memoryless discrete iteration which can then be approximated by a continuous trajectory. Background on the method of modified equations can be found in Ernst Hairer and Wanner [23], Li et al. [42].

Works deriving modified equations for (S)GD with or without momentum include Barrett and Dherin [6], Smith et al. [59], Ghosh et al. [26], Farazmand [24], Kovachki and Stuart [37], Miyagawa [50], Rosca et al. [56], Li et al. [42]. In particular, Ghosh et al. [26] identified that momentum strongly regularizes the loss function in the case of GD, though their error bounds both have a different focus (continuous approximation rather than discrete one), and follow a different approach which appears hard to generalize to other algorithms. Our approach works for a wide class of algorithms, and we recover their main results in Section 5. Works approximating adaptive methods with continuous trajectories include Ma et al. [47], Malladi et al. [48], Barakat and Bianchi [5], Compagnoni et al. [17]. More recently, Cattaneo et al. [11] studied the special case of Adam / RMSProp. We built on this work to conduct empirical evaluations. Their focus is not on memory but on continuous approximations; in particular, they do not have approximation bounds between two discrete iterations like we do. In addition, their arguments are highly specialized to Adam, and they do not incorporate weight decay. Although we also discuss Adam (with weight decay) extensively, it is only because of its importance in practice, and our results cover a much broader class of optimizers.

This paper is also connected to the strand of the literature studying the implicit bias of optimization algorithms. For example, Xie and Li [66] and Chen et al. [12] prove that weight decay causes AdamW and Lion to solve an $\ell_\infty$-norm constrained optimization problem. In that, they behave asymptotically like (normalized) steepest descent with respect to $\ell_\infty$-norm. Bernstein and Newhouse [7] also view Adam and Lion as smoothed-out versions of steepest descent. This perspective is connected to the Moreau-Yosida approximation of the loss function [9]; the latter work provides a concrete way to write down popular optimizers (including SGD with momentum, RMSProp and Adam) as a sequence of optimization problems. In addition, a large body of work is devoted to the bias of optimization algorithms towards the direction of the max-margin vector [60, 51, 52, 55, 63, 29, 32, 33]. Similarly, Damian et al. [19], Li et al. [44], Arora et al. [4], Wen et al. [64], Damian et al. [20] explore the sharpness of regions SGD converges to. Gunasekar et al. [28], Arora et al. [3] study implicit regularization in matrix factorization. Li et al. [43] prove in a certain setting that a larger learning rate leads to better generalization.

# B    Broader Impacts

This paper presents a general framework for contrasting certain properties of optimization algorithms commonly used for training neural networks, and thus this work can lead to societal consequences as common of deep learning.

# C    Special Case: $\boldsymbol{F}^{(n)}$ as a Function of "Momentum Variables"

In the examples listed in the introduction, $\boldsymbol{F}^{(n)}$ satisfies a more specific form that can be used to give more primitive conditions for Assumption 3.1. The following assumption, which is a special case of Assumption 3.1 by Lemma H.1, may look a bit technical but allows for a simpler calculation of correction terms.

**Assumption C.1** (Special case of Assumption 3.1: $\boldsymbol{F}^{(n)}$ is a function of "momentum variables"). *Let $\{\boldsymbol{g}_\ell^{(n)}\}_{\ell=1}^L$ be $L$ two times continuously differentiable functions $\mathcal{D} \to \mathbb{R}^d$, uniformly bounded along with two derivatives. Let $\{\beta_\ell\}_{\ell=1}^L$ be fixed reals in $[0, 1)$, and $\{b_\ell^{(n+1)}\}_{\ell=1}^L$ be $L$ bounded nonnegative*

sequences of reals (for $n \in \mathbb{Z}_{\geqslant 0}$). Assume the function $\boldsymbol{F}^{(n)}$ has the form

$$\boldsymbol{F}^{(n)}(\boldsymbol{\theta}^{(n)}, \ldots, \boldsymbol{\theta}^{(0)}) := \boldsymbol{\Phi}(\boldsymbol{m}_1^{(n+1)}, \ldots, \boldsymbol{m}_L^{(n+1)})$$

$$\text{with } \boldsymbol{m}_\ell^{(n+1)} := b_\ell^{(n+1)} \sum_{k=0}^{n} \beta_\ell^k \boldsymbol{g}_\ell^{(n-k)}(\boldsymbol{\theta}^{(n-k)}) \in \mathcal{M}, \tag{11}$$

where $\mathcal{M}$ is a bounded open region in $\mathbb{R}^d$ and $\boldsymbol{\Phi}(\boldsymbol{m}_1, \ldots, \boldsymbol{m}_L) \colon \mathcal{M}^L \to \mathbb{R}^d$ is a fixed two times continuously differentiable function, uniformly bounded along with two derivatives. In the full-batch case, $\boldsymbol{g}_\ell^{(n)} \equiv \boldsymbol{g}_\ell$ are not allowed to depend on $n$.

For instance, in the case of AdamW (Example 1.3), Assumption C.1 applies with $L = 3$,

$$\boldsymbol{g}_1(\boldsymbol{\theta}) = \nabla \mathcal{L}(\boldsymbol{\theta}), \quad \boldsymbol{g}_2(\boldsymbol{\theta}) = \big(\nabla \mathcal{L}(\boldsymbol{\theta})\big)^2, \quad \boldsymbol{g}_3(\boldsymbol{\theta}) = \boldsymbol{\theta},$$

$$b_1^{(n+1)} = \frac{1 - \beta_1}{1 - \beta_1^{n+1}} \to b_1 = 1 - \beta_1,$$

$$b_2^{(n+1)} = \frac{1 - \beta_2}{1 - \beta_2^{n+1}} \to b_2 = 1 - \beta_2,$$

$$b_3^{(n+1)} \equiv b_3 = \lambda, \quad \beta_3 = 0.$$

In the case of Lion-$\mathcal{K}$ (Example 1.5), the assumption applies with $L = 3$,

$$\boldsymbol{g}_1(\boldsymbol{\theta}) = -\nabla \mathcal{L}(\boldsymbol{\theta}), \quad \boldsymbol{g}_2(\boldsymbol{\theta}) = -\nabla \mathcal{L}(\boldsymbol{\theta}), \quad \boldsymbol{g}_3(\boldsymbol{\theta}) = \boldsymbol{\theta},$$

$$\beta_1 = \rho_2, \quad \beta_2 = 0, \quad \beta_3 = 0,$$

$$b_1^{(n+1)} \equiv b_1 = (1 - \rho_2)\frac{\rho_1}{\rho_2},$$

$$b_2^{(n+1)} \equiv b_2 = 1 - \frac{\rho_1}{\rho_2},$$

$$b_3^{(n+1)} \equiv b_3 = \lambda.$$

*We used the letter $\rho$ when defining the Lion iteration to avoid confusion with the $\beta$ in the definition of "momentum variables".*

Specializing to the setup of Assumption C.1, and for any $s, n \in \mathbb{Z}_{\geqslant 0}$, $\boldsymbol{F}^{(s)}(\boldsymbol{\theta}) = \boldsymbol{\Phi}\big(\bar{\boldsymbol{g}}_1^{(s+1)}(\boldsymbol{\theta}), \ldots, \bar{\boldsymbol{g}}_L^{(s+1)}(\boldsymbol{\theta})\big)$, where $\bar{\boldsymbol{g}}_\ell^{(s+1)}(\boldsymbol{\theta}) := b_\ell^{(n+1)} \sum_{k=0}^{s} \beta_\ell^k \boldsymbol{g}_\ell^{(n-k)}(\boldsymbol{\theta})$, and

$$\frac{\partial F_r^{(n)}}{\partial \boldsymbol{\theta}^{(n-k)}}(\boldsymbol{\theta}) = \sum_{\ell=1}^{L} \sum_i b_\ell^{(n+1)} \beta_\ell^k \frac{\partial \Phi_r}{\partial m_{\ell;i}} \big(\bar{\boldsymbol{g}}_1^{(n+1)}(\boldsymbol{\theta}), \ldots, \bar{\boldsymbol{g}}_L^{(n+1)}(\boldsymbol{\theta})\big)^\mathsf{T} \nabla g_{\ell;i}^{(n-k)}(\boldsymbol{\theta}).$$

Therefore, in this special case, the correction term in the memoryless iteration (4) is given by, for $r = 1, \ldots, d$,

$$M_r^{(n)}(\boldsymbol{\theta}) = h \sum_{\ell=1}^{L} \sum_i b_\ell^{(n+1)} \frac{\partial \Phi_r}{\partial m_{\ell;i}} \big(\bar{\boldsymbol{g}}_1^{(n+1)}(\boldsymbol{\theta}), \ldots, \bar{\boldsymbol{g}}_L^{(n+1)}(\boldsymbol{\theta})\big) \times$$

$$\times \sum_{k=1}^{n} \beta_\ell^k \nabla g_{\ell;i}^{(n-k)}(\boldsymbol{\theta})^\mathsf{T} \sum_{s=n-k}^{n-1} \boldsymbol{\Phi}\big(\bar{\boldsymbol{g}}_1^{(s+1)}(\boldsymbol{\theta}), \ldots, \bar{\boldsymbol{g}}_L^{(s+1)}(\boldsymbol{\theta})\big).$$

In the full-batch case $\boldsymbol{g}_\ell^{(n)}(\boldsymbol{\theta}) \equiv \boldsymbol{g}_\ell(\boldsymbol{\theta})$, this can be simplified further. Let us assume $b_\ell^{(n+1)} \xrightarrow[n \to \infty]{} b_\ell$, where $b_\ell$ is constant in $n$. Then $\bar{\boldsymbol{g}}_\ell^{(n+1)}(\boldsymbol{\theta})$ also become constant in $n$: specifically, they settle to $\bar{\boldsymbol{g}}_\ell(\boldsymbol{\theta}) := b_\ell (1 - \beta_\ell)^{-1} \boldsymbol{g}_\ell(\boldsymbol{\theta})$. Lemma H.2 then implies that the iteration becomes close to

$$\boldsymbol{\theta}^{(n+1)} = \boldsymbol{\theta}^{(n)} - h\big[\boldsymbol{\Phi}(\bar{\boldsymbol{g}}_1(\boldsymbol{\theta}^{(n)}), \ldots, \bar{\boldsymbol{g}}_L(\boldsymbol{\theta}^{(n)})) + \boldsymbol{M}^{(n)}(\boldsymbol{\theta})\big]$$

with

$$M_r^{(n)}(\boldsymbol{\theta}) = h \sum_{\ell=1}^{L} \sum_i \frac{b_\ell \beta_\ell}{(1 - \beta_\ell)^2} \frac{\partial \Phi_r}{\partial m_{\ell;i}} (\bar{\boldsymbol{g}}_1(\boldsymbol{\theta}^{(n)}), \ldots, \bar{\boldsymbol{g}}_L(\boldsymbol{\theta}^{(n)})) \times$$

$$\times \nabla g_{\ell;i}(\boldsymbol{\theta}^{(n)})^{\mathsf{T}} \boldsymbol{\Phi}(\bar{\boldsymbol{g}}_1(\boldsymbol{\theta}^{(n)}), \ldots, \bar{\boldsymbol{g}}_L(\boldsymbol{\theta}^{(n)})).$$

These formulae admittedly look complicated, but we can easily plug in the definitions and calculate correction terms for all examples with little additional algebra. We list these terms in Section D.

# D Details for Examples: Correction Terms

For GD with momentum (Example 1.1):

$$\boldsymbol{M}^{(n)}(\boldsymbol{\theta}) = \frac{h\beta}{2(1 - \beta)^3} \nabla \|\nabla \mathcal{L}(\boldsymbol{\theta})\|^2.$$

For Nesterov's accelerated GD (Example 1.2):

$$\boldsymbol{M}^{(n)}(\boldsymbol{\theta}) = \frac{h\beta^2}{2(1 - \beta)^3} \nabla \|\nabla \mathcal{L}(\boldsymbol{\theta})\|^2.$$

For AdamW (Example 1.3, also discussed in Section 4):

$$\boldsymbol{M}^{(n)}(\boldsymbol{\theta}) = h \left( \frac{\beta_1(1 - \beta_1)^{-1} - \beta_2(1 - \beta_2)^{-1}}{(|\nabla \mathcal{L}(\boldsymbol{\theta})|^2 + \varepsilon)^{1/2}} + \varepsilon \frac{\beta_2(1 - \beta_2)^{-1}}{(|\nabla \mathcal{L}(\boldsymbol{\theta})|^2 + \varepsilon)^{3/2}} \right) \left( \nabla \|\nabla \mathcal{L}(\boldsymbol{\theta})\|_{1,\varepsilon} + \lambda \nabla^2 \mathcal{L}(\boldsymbol{\theta}) \boldsymbol{\theta} \right).$$

For Nadam (Example 1.4):

$$\boldsymbol{M}^{(n)}(\boldsymbol{\theta}) = h \left( \frac{\beta_1^2(1 - \beta_1)^{-1} - \beta_2(1 - \beta_2)^{-1}}{(|\nabla \mathcal{L}(\boldsymbol{\theta})|^2 + \varepsilon)^{1/2}} + \varepsilon \frac{\beta_2(1 - \beta_2)^{-1}}{(|\nabla \mathcal{L}(\boldsymbol{\theta})|^2 + \varepsilon)^{3/2}} \right) \left( \nabla \|\nabla \mathcal{L}(\boldsymbol{\theta})\|_{1,\varepsilon} + \lambda \nabla^2 \mathcal{L}(\boldsymbol{\theta}) \boldsymbol{\theta} \right).$$

For Lion-$\mathcal{K}$ (Example 1.5, also discussed in Section 4):

$$\boldsymbol{M}^{(n)}(\boldsymbol{\theta}) = -h \frac{\rho_1}{1 - \rho_2} \nabla^2 \mathcal{K}(-\nabla \mathcal{L}(\boldsymbol{\theta})) \nabla^2 \mathcal{L}(\boldsymbol{\theta}) \left[ \nabla \mathcal{K}(-\nabla \mathcal{L}(\boldsymbol{\theta})) - \lambda \boldsymbol{\theta} \right].$$

# E Mini-Batch Training: Details

Let us take the expectation of the correction term with respect to the random permutation of mini-batches, that is, take the average over all re-orderings $\pi$ of $(\boldsymbol{g}^{(0)}, \ldots, \boldsymbol{g}^{(n)})$:

$$\mathbb{E} \sum_{k=0}^{n-1} \beta^k \nabla g_r^{(\pi(n-1-k))}(\boldsymbol{\theta})^{\mathsf{T}} \sum_{l=1}^{k+1} \sum_{b=0}^{n-l} \beta^b \boldsymbol{g}^{(\pi(n-l-b))}(\boldsymbol{\theta})$$

$$:= \frac{1}{(n+1)!} \sum_{\pi} \sum_{k=0}^{n-1} \beta^k \nabla g_r^{(\pi(n-1-k))}(\boldsymbol{\theta})^{\mathsf{T}} \sum_{l=1}^{k+1} \sum_{b=0}^{n-l} \beta^b \boldsymbol{g}^{(\pi(n-l-b))}(\boldsymbol{\theta}).$$

Note that $\mathbb{E} \nabla g_r^{(i)}(\boldsymbol{\theta})^{\mathsf{T}} \boldsymbol{g}^{(j)}(\boldsymbol{\theta})$ depends only on whether $i = j$ or $i \neq j$. Therefore,

$$\mathbb{E}[M_r^{(n)}(\boldsymbol{\theta})]/h = C_3(\beta) \mathbb{E}[\nabla g_r^{(1)}(\boldsymbol{\theta})^{\mathsf{T}} \boldsymbol{g}^{(1)}(\boldsymbol{\theta})] + C_4(\beta) \mathbb{E}[\nabla g_r^{(1)}(\boldsymbol{\theta})^{\mathsf{T}} \boldsymbol{g}^{(2)}(\boldsymbol{\theta})],$$

where $C_3(\beta)$ and $C_4(\beta)$ are given by

$$C_3(\beta) := \beta \sum_{b=0}^{n-1} \beta^b \sum_{l=1}^{b+1} \beta^{b+1-l} \xrightarrow[n\to\infty]{} \frac{\beta}{(1 - \beta)^2(1 + \beta)},$$

$$C_4(\beta) := \beta \sum_{k=0}^{n-1} \beta^k \sum_{l=1}^{k+1} \sum_{b=0}^{n-l} \beta^b - C_3(\beta) \xrightarrow[n\to\infty]{} \frac{2\beta^2}{(1 - \beta)^3(1 + \beta)}.$$

We can simplify

$$\mathbb{E}[\nabla g_r^{(1)}(\boldsymbol{\theta})^\mathsf{T} \boldsymbol{g}^{(2)}(\boldsymbol{\theta})] = \frac{1}{(n+1)n} \sum_{i \neq j} \nabla g_r^{(i)}(\boldsymbol{\theta})^\mathsf{T} \boldsymbol{g}^{(j)}(\boldsymbol{\theta}) = \nabla g_r(\boldsymbol{\theta})^\mathsf{T} \boldsymbol{g}(\boldsymbol{\theta}) + o_n(1),$$

where $\boldsymbol{g}(\boldsymbol{\theta}) = \mathbb{E}\boldsymbol{g}^{(1)}(\boldsymbol{\theta}) = (n+1)^{-1} \sum_{k=0}^{n+1} \boldsymbol{g}^{(k)}(\boldsymbol{\theta})$ is the average of $\{\boldsymbol{g}^{(k)}(\boldsymbol{\theta})\}$, $o_n(1)$ tends to zero as $n \to \infty$.

So, for large $n$ we can write

$$\mathbb{E}[M_r^{(n)}(\boldsymbol{\theta})]/h \approx C_3(\beta)\mathbb{E}[\nabla g_r^{(1)}(\boldsymbol{\theta})^\mathsf{T} \boldsymbol{g}^{(1)}(\boldsymbol{\theta})] + C_4(\beta)\nabla g_r(\boldsymbol{\theta})^\mathsf{T} \boldsymbol{g}(\boldsymbol{\theta})$$
$$= \left(C_3(\beta) + C_4(\beta)\right)\nabla g_r(\boldsymbol{\theta})^\mathsf{T} \boldsymbol{g}(\boldsymbol{\theta}) + C_3(\beta)\mathbb{E}[(\nabla g_r^{(1)}(\boldsymbol{\theta}) - \nabla g_r(\boldsymbol{\theta}))^\mathsf{T} (\boldsymbol{g}^{(1)}(\boldsymbol{\theta}) - \boldsymbol{g}(\boldsymbol{\theta}))]$$
$$\approx \frac{\beta}{(1-\beta)^3}\nabla g_r(\boldsymbol{\theta})^\mathsf{T} \boldsymbol{g}(\boldsymbol{\theta}) + C_3(\beta)\mathbb{E}[(\nabla g_r^{(1)}(\boldsymbol{\theta}) - \nabla g_r(\boldsymbol{\theta}))^\mathsf{T} (\boldsymbol{g}^{(1)}(\boldsymbol{\theta}) - \boldsymbol{g}(\boldsymbol{\theta}))].$$

# F   Proof of Theorem 3.2

Since, by the assumptions of the theorem,

$$\tilde{\theta}_r^{(n+1)} - \tilde{\theta}_r^{(n)} = -hF_r^{(n)}(\tilde{\boldsymbol{\theta}}^{(n)}, \ldots, \tilde{\boldsymbol{\theta}}^{(n)})$$
$$- h^2 \sum_{k=1}^{n} \frac{\partial F_r^{(n)}}{\partial \boldsymbol{\theta}^{(n-k)}}(\tilde{\boldsymbol{\theta}}^{(n)}, \ldots, \tilde{\boldsymbol{\theta}}^{(n)})^\mathsf{T} \underline{\sum_{s=n-k}^{n-1} \boldsymbol{F}^{(s)}(\tilde{\boldsymbol{\theta}}^{(n)}, \ldots, \tilde{\boldsymbol{\theta}}^{(n)})}, \tag{12}$$

we need to show that

$$F_r^{(n)}(\tilde{\boldsymbol{\theta}}^{(n)}, \ldots, \tilde{\boldsymbol{\theta}}^{(0)}) - F_r^{(n)}(\tilde{\boldsymbol{\theta}}^{(n)}, \ldots, \tilde{\boldsymbol{\theta}}^{(n)})$$
$$= h \sum_{k=1}^{n} \frac{\partial F_r^{(n)}}{\partial \boldsymbol{\theta}^{(n-k)}}(\tilde{\boldsymbol{\theta}}^{(n)}, \ldots, \tilde{\boldsymbol{\theta}}^{(n)})^\mathsf{T} \sum_{s=n-k}^{n-1} \boldsymbol{F}^{(s)}(\tilde{\boldsymbol{\theta}}^{(n)}, \ldots, \tilde{\boldsymbol{\theta}}^{(n)}) + O(h^2). \tag{13}$$

By Taylor expansion with the Lagrange remainder,

$$F_r^{(n)}(\tilde{\boldsymbol{\theta}}^{(n)}, \ldots, \tilde{\boldsymbol{\theta}}^{(0)}) - F_r^{(n)}(\tilde{\boldsymbol{\theta}}^{(n)}, \ldots, \tilde{\boldsymbol{\theta}}^{(n)})$$
$$= \sum_{k=1}^{n} \frac{\partial F_r^{(n)}}{\partial \boldsymbol{\theta}^{(n-k)}}(\tilde{\boldsymbol{\theta}}^{(n)}, \ldots, \tilde{\boldsymbol{\theta}}^{(n)})^\mathsf{T} (\tilde{\boldsymbol{\theta}}^{(n-k)} - \tilde{\boldsymbol{\theta}}^{(n)})$$
$$+ \frac{1}{2} \sum_{k_1,k_2=1}^{n} (\tilde{\boldsymbol{\theta}}^{(n-k_1)} - \tilde{\boldsymbol{\theta}}^{(n)})^\mathsf{T} \frac{\partial^2 F_r^{(n)}}{\partial \boldsymbol{\theta}^{(n-k_1)} \partial \boldsymbol{\theta}^{(n-k_2)}}(\boldsymbol{\zeta})(\tilde{\boldsymbol{\theta}}^{(n-k_2)} - \tilde{\boldsymbol{\theta}}^{(n)})$$
$$\overset{(a)}{=} \sum_{k=1}^{n} \frac{\partial F_r^{(n)}}{\partial \boldsymbol{\theta}^{(n-k)}}(\tilde{\boldsymbol{\theta}}^{(n)}, \ldots, \tilde{\boldsymbol{\theta}}^{(n)})^\mathsf{T} (\tilde{\boldsymbol{\theta}}^{(n-k)} - \tilde{\boldsymbol{\theta}}^{(n)}) + O(h^2), \tag{14}$$

where $\boldsymbol{\zeta}$ is some point on the segment between $(\tilde{\boldsymbol{\theta}}^{(n)}, \ldots, \tilde{\boldsymbol{\theta}}^{(0)})$ and $(\tilde{\boldsymbol{\theta}}^{(n)}, \ldots, \tilde{\boldsymbol{\theta}}^{(n)})$; in (a) we used Assumption 3.1 and $\tilde{\boldsymbol{\theta}}^{(n-k_1)} - \tilde{\boldsymbol{\theta}}^{(n)} = O(k_1 h)$, $\tilde{\boldsymbol{\theta}}^{(n-k_2)} - \tilde{\boldsymbol{\theta}}^{(n)} = O(k_2 h)$. Since the underlined term in Equation (12) is $O(1)$, we have

$$\tilde{\boldsymbol{\theta}}^{(n-k)} - \tilde{\boldsymbol{\theta}}^{(n)} = \sum_{s=n-k}^{n-1} (\tilde{\boldsymbol{\theta}}^{(s)} - \tilde{\boldsymbol{\theta}}^{(s+1)})$$
$$= h \sum_{s=n-k}^{n-1} \left\{ \boldsymbol{F}^{(s)}(\tilde{\boldsymbol{\theta}}^{(s)}, \ldots, \tilde{\boldsymbol{\theta}}^{(s)}) + O(h) \right\}$$
$$= h \sum_{s=n-k}^{n-1} \boldsymbol{F}^{(s)}(\tilde{\boldsymbol{\theta}}^{(n)}, \ldots, \tilde{\boldsymbol{\theta}}^{(n)}) + O(k^2 h^2),$$

where in the last step we used $\boldsymbol{F}^{(s)}(\tilde{\boldsymbol{\theta}}^{(s)}, \ldots, \tilde{\boldsymbol{\theta}}^{(s)}) - \boldsymbol{F}^{(s)}(\tilde{\boldsymbol{\theta}}^{(n)}, \ldots, \tilde{\boldsymbol{\theta}}^{(n)}) = O((n-s)h)$, which follows from Taylor expansion, Assumption 3.1 and $\tilde{\boldsymbol{\theta}}^{(n+1)} - \tilde{\boldsymbol{\theta}}^{(n)} = O(h)$. Combine this with Equation (14) to get

$$
F_r^{(n)}(\tilde{\boldsymbol{\theta}}^{(n)}, \ldots, \tilde{\boldsymbol{\theta}}^{(0)}) - F_r^{(n)}(\tilde{\boldsymbol{\theta}}^{(n)}, \ldots, \tilde{\boldsymbol{\theta}}^{(n)})
$$

$$
= \sum_{k=1}^{n} \frac{\partial F_r^{(n)}}{\partial \boldsymbol{\theta}^{(n-k)}}(\tilde{\boldsymbol{\theta}}^{(n)}, \ldots, \tilde{\boldsymbol{\theta}}^{(n)})^{\mathsf{T}} \left\{ h \sum_{s=n-k}^{n-1} \boldsymbol{F}^{(s)}(\tilde{\boldsymbol{\theta}}^{(n)}, \ldots, \tilde{\boldsymbol{\theta}}^{(n)}) + O(k^2 h^2) \right\} + O(h^2)
$$

$$
= h \sum_{k=1}^{n} \frac{\partial F_r^{(n)}}{\partial \boldsymbol{\theta}^{(n-k)}}(\tilde{\boldsymbol{\theta}}^{(n)}, \ldots, \tilde{\boldsymbol{\theta}}^{(n)})^{\mathsf{T}} \sum_{s=n-k}^{n-1} \boldsymbol{F}^{(s)}(\tilde{\boldsymbol{\theta}}^{(n)}, \ldots, \tilde{\boldsymbol{\theta}}^{(n)}) + O(h^2),
$$

which is (13), and the proof is complete.

## G  Proof of Corollary 3.3

We follow a standard argument, e. g. Ghosh et al. [26], Cattaneo et al. [11]. We prove the following claim by induction over $n \in \mathbb{Z}_{\geqslant 0}$:

$$
\|\boldsymbol{\theta}^{(n)} - \tilde{\boldsymbol{\theta}}^{(n)}\|_\infty \leqslant d_1 e^{d_2 n h} h^2, \quad \|\boldsymbol{\theta}^{(n+1)} - \tilde{\boldsymbol{\theta}}^{(n+1)} - \boldsymbol{\theta}^{(n)} + \tilde{\boldsymbol{\theta}}^{(n)}\|_\infty \leqslant d_3 e^{d_2 n h} h^3,
$$

where

$$
d_1 = C_1, \quad d_2 = 1 + d \sum_{k=0}^{\infty} \gamma_k, \quad d_3 = C_1 d_2.
$$

Because $nh \leqslant T$, Corollary 3.3 will follow.

Base: $n = 0$. It is indeed true that $\|\boldsymbol{\theta}^{(0)} - \tilde{\boldsymbol{\theta}}^{(0)}\|_\infty \leqslant d_1 h^2$ because the left-hand side is zero. It is indeed true that $\|\boldsymbol{\theta}^{(1)} - \tilde{\boldsymbol{\theta}}^{(1)} - \boldsymbol{\theta}^{(0)} + \tilde{\boldsymbol{\theta}}^{(0)}\|_\infty \leqslant d_3 h^3$ for the same reason.

Assume $n \in \mathbb{Z}_{\geqslant 1}$ and it is true that

$$
\|\boldsymbol{\theta}^{(n')} - \tilde{\boldsymbol{\theta}}^{(n')}\|_\infty \leqslant d_1 e^{d_2 n' h} h^2, \quad \|\boldsymbol{\theta}^{(n'+1)} - \tilde{\boldsymbol{\theta}}^{(n'+1)} - \boldsymbol{\theta}^{(n')} + \tilde{\boldsymbol{\theta}}^{(n')}\|_\infty \leqslant d_3 e^{d_2 n' h} h^3.
$$

for all $0 \leqslant n' \leqslant n - 1$. Then

$$
\|\boldsymbol{\theta}^{(n)} - \tilde{\boldsymbol{\theta}}^{(n)}\|_\infty \leqslant \|\boldsymbol{\theta}^{(n-1)} - \tilde{\boldsymbol{\theta}}^{(n-1)}\|_\infty + \|\boldsymbol{\theta}^{(n)} - \tilde{\boldsymbol{\theta}}^{(n)} - \boldsymbol{\theta}^{(n-1)} + \tilde{\boldsymbol{\theta}}^{(n-1)}\|_\infty
$$

by the triangle inequality,

$$
\leqslant d_1 e^{d_2(n-1)h} h^2 + d_3 e^{d_2(n-1)h} h^3
$$

by the induction hypothesis,

$$
= d_1 \left( 1 + \frac{d_3}{d_1} h \right) e^{d_2(n-1)h} h^2 \leqslant d_1 (1 + d_2 h) e^{d_2(n-1)h} h^2
$$

by $d_3 \leqslant d_1 d_2$,

$$
\leqslant d_1 e^{d_2 n h} h^2
$$

by the inequality $1 + x \leqslant e^x$ for all $x \geqslant 0$.

Next, write

$$
\boldsymbol{\theta}^{(n+1)} - \boldsymbol{\theta}^{(n)} - \tilde{\boldsymbol{\theta}}^{(n+1)} + \tilde{\boldsymbol{\theta}}^{(n)}
$$

$$
= -h \boldsymbol{F}^{(n)}(\boldsymbol{\theta}^{(n)}, \ldots, \boldsymbol{\theta}^{(0)}) - \{\tilde{\boldsymbol{\theta}}^{(n+1)} - \tilde{\boldsymbol{\theta}}^{(n)}\}
$$

$$= h\big[\boldsymbol{F}^{(n)}(\tilde{\boldsymbol{\theta}}^{(n)},\ldots,\tilde{\boldsymbol{\theta}}^{(0)}) - \boldsymbol{F}^{(n)}(\boldsymbol{\theta}^{(n)},\ldots,\boldsymbol{\theta}^{(0)})\big] - \{\tilde{\boldsymbol{\theta}}^{(n+1)} - \tilde{\boldsymbol{\theta}}^{(n)} + h\boldsymbol{F}^{(n)}(\tilde{\boldsymbol{\theta}}^{(n)},\ldots,\tilde{\boldsymbol{\theta}}^{(0)})\}$$

Then

$$|\theta_r^{(n+1)} - \theta_r^{(n)} - \tilde{\theta}_r^{(n+1)} + \tilde{\theta}_r^{(n)}|$$

$$\leqslant h\big|F_r^{(n)}(\tilde{\boldsymbol{\theta}}^{(n)},\ldots,\tilde{\boldsymbol{\theta}}^{(0)}) - F_r^{(n)}(\boldsymbol{\theta}^{(n)},\ldots,\boldsymbol{\theta}^{(0)})\big| + \big|\tilde{\theta}_r^{(n+1)} - \tilde{\theta}_r^{(n)} + hF_r^{(n)}(\tilde{\boldsymbol{\theta}}^{(n)},\ldots,\tilde{\boldsymbol{\theta}}^{(0)})\big|$$

$$\leqslant h\big|F_r^{(n)}(\tilde{\boldsymbol{\theta}}^{(n)},\ldots,\tilde{\boldsymbol{\theta}}^{(0)}) - F_r^{(n)}(\boldsymbol{\theta}^{(n)},\ldots,\boldsymbol{\theta}^{(0)})\big| + C_1 h^3$$

by Theorem 3.2,

$$= h\left|\sum_{k=0}^{n}\frac{\partial F_r^{(n)}}{\partial\boldsymbol{\theta}^{(n-k)}}(\boldsymbol{\zeta})^\mathsf{T}(\tilde{\boldsymbol{\theta}}^{(n-k)} - \boldsymbol{\theta}^{(n-k)})\right| + C_1 h^3,$$

where $\boldsymbol{\zeta}$ is a point on the segment between $(\tilde{\boldsymbol{\theta}}^{(n)},\ldots,\tilde{\boldsymbol{\theta}}^{(0)})$ and $(\boldsymbol{\theta}^{(n)},\ldots,\boldsymbol{\theta}^{(0)})$,

$$\leqslant hd\sum_{k=0}^{n}\gamma_k\|\tilde{\boldsymbol{\theta}}^{(n-k)} - \boldsymbol{\theta}^{(n-k)}\|_\infty + C_1 h^3$$

by (3.1) (recall that $d$ is the dimension of $\boldsymbol{\theta}$),

$$\leqslant d_1 h^3 d\sum_{k=0}^{\infty}\gamma_k e^{d_2(n-k)h} + C_1 h^3$$

by the induction hypothesis and the bound on $\|\tilde{\boldsymbol{\theta}}^{(n)} - \boldsymbol{\theta}^{(n)}\|_\infty$ already proven

$$\leqslant \underbrace{\left(d_1 d\sum_{k=0}^{\infty}\gamma_k + C_1\right)}_{\leqslant d_3} e^{d_2 nh} h^3$$

$$\leqslant d_3 e^{d_2 nh} h^3.$$

## H   Auxiliary Results

**Lemma H.1** (Memory decays exponentially fast). *If $\boldsymbol{F}^{(n)}$ is a function of "momentum variables" as described in Assumption C.1, then for any $n$ and $k \leqslant n$*

$$\max_{r,i}\left|\frac{\partial F_r^{(n)}}{\partial\theta_i^{(n-k)}}\right| \leqslant \gamma_k, \tag{15}$$

*and similarly for any $n$ and $k_1, k_2 \leqslant n$*

$$\max_{r,i,j}\left|\frac{\partial^2 F_r^{(n)}}{\partial\theta_i^{(n-k_1)}\partial\theta_j^{(n-k_2)}}\right| \leqslant \gamma_{k_1,k_2}, \tag{16}$$

*where $\{\gamma_k\}$ and $\{\gamma_{k_1,k_2}\}$ are sequences decaying exponentially fast: specifically,*

$$\gamma_k := C_\gamma\{\max_\ell \beta_\ell\}^k, \quad \gamma_{k_1,k_2} := C_\gamma\{\max_\ell \beta_\ell\}^{k_1+k_2}$$

*for some constant $C_\gamma > 0$.*

*Proof.* It is easy to see (15) by taking the derivative:

$$\frac{\partial}{\partial\boldsymbol{\theta}^{(n-k)}}F_r^{(n)}(\boldsymbol{\theta}^{(n)},\ldots,\boldsymbol{\theta}^{(0)})$$

$$= \sum_{\ell=1}^{L} \sum_{i} \frac{\partial \Phi_r}{\partial m_{\ell;i}}(\boldsymbol{m}_1^{(n+1)}, \ldots, \boldsymbol{m}_L^{(n+1)}) \frac{\partial m_{\ell;i}^{(n+1)}}{\boldsymbol{\theta}^{(n-k)}}$$

$$= \sum_{\ell=1}^{L} \sum_{i} b_\ell^{(n+1)} \beta_\ell^k \frac{\partial \Phi_r}{\partial m_{\ell;i}}(\boldsymbol{m}_1^{(n+1)}, \ldots, \boldsymbol{m}_L^{(n+1)})$$

$$\times \nabla g_{\ell;i}^{(n-k)}(\boldsymbol{\theta}^{(n-k)}),$$

and using the uniform boundedness of derivatives of $g_{\ell;i}^{(n-k)}$ and $\Phi_r$. Equation (16) is proven similarly. $\qquad\square$

**Lemma H.2.** *Let $\{a_k\}_{k=1}^{\infty}$ and $\{b_k\}_{k=1}^{\infty}$ be sequences of reals such that $\sum_{k=1}^{\infty}(|a_k| + |b_k|) < \infty$. Then*

$$\sum_{k=1}^{n} a_k \sum_{s=n-k}^{n-1} b_s \xrightarrow[n\to\infty]{} 0.$$

*Proof.* Fix $\varepsilon > 0$. Take such positive integer $n_0$ that for any $n_0 \leqslant n_1 \leqslant n_2$ we have $\sum_{s=n_1}^{n_2}(|a_k| + |b_k|) < \varepsilon$. Then for any $n \geqslant 2n_0 - 1$ the following holds:

$$\sum_{k=1}^{n}|a_k| \sum_{s=n-k}^{n-1}|b_s| = \sum_{k=1}^{n-n_0}|a_k| \underbrace{\sum_{s=n-k}^{n-1}|b_s|}_{<\varepsilon} + \underbrace{\sum_{k=n-n_0+1}^{n}|a_k|}_{<\varepsilon} \sum_{s=n-k}^{n-1}|b_s| < \varepsilon \sum_{k=1}^{\infty}(|a_k| + |b_k|).$$

Since $\varepsilon$ is arbitrary and $\sum_{k=1}^{\infty}(|a_k| + |b_k|)$ is a finite constant, the statement follows. $\qquad\square$

# I   Corollaries for Special Cases

**Lemma I.1** (Application of Corollary 3.3 to Example 1.1)**.** *Let $\{\boldsymbol{\theta}^{(n)}\}_{n\in\mathbb{Z}_{\geqslant 0}}$ be the sequence of vectors generated by the iteration in Equation (1) with initial condition $\boldsymbol{\theta}^{(0)}$, where $\boldsymbol{F}^{(n)}(\cdot)$ is as defined in Example 1.1, and the loss function $\mathcal{L}(\cdot)$ defined in an open convex bounded domain $\mathcal{D} \subset \mathbb{R}^d$ is three times continuously differentiable with bounded derivatives; also, let $T \geqslant 0$ be a fixed "time" horizon. Then Assumption 3.1 holds; in particular, by Corollary 3.3 the inequality $\max_{n\in[0:\lfloor T/h\rfloor]}\left\|\boldsymbol{\theta}^{(n)} - \tilde{\boldsymbol{\theta}}^{(n)}\right\|_\infty \leqslant C_2 h^2$ holds, where*

$$\tilde{\theta}_r^{(n+1)} = \tilde{\theta}_r^{(n)} - h\left(\frac{1 - \beta^{n+1}}{1 - \beta}\nabla_r\mathcal{L}(\tilde{\boldsymbol{\theta}}^{(n)}) + M_r^{(n)}(\tilde{\boldsymbol{\theta}}^{(n)})\right),$$

$$M_r^{(n)}(\boldsymbol{\theta}) = h\frac{\beta[1 - (2n + 1)\beta^n(1 - \beta) - \beta^{2n+1}]}{2(1 - \beta)^3}\nabla_r\|\nabla\mathcal{L}(\boldsymbol{\theta})\|^2, \quad r \in [1:d].$$

*Proof.* The fact that Assumption 3.1 holds is already verified in Section C.

Next, in this case

$$F_r^{(n)}(\boldsymbol{\theta}) = \frac{1 - \beta^{n+1}}{1 - \beta}\nabla_r\mathcal{L}(\boldsymbol{\theta}),$$

$$\frac{\partial F_r^{(n)}}{\partial \theta_i^{(n-k)}}(\boldsymbol{\theta}) = \beta^k \nabla_{ir}\mathcal{L}(\boldsymbol{\theta}).$$

Therefore,

$$M_r^{(n)}\boldsymbol{\theta} = h\sum_{k=1}^{n}\sum_{i=1}^{d}\frac{\partial F_r^{(n)}}{\partial \theta_i^{(n-k)}}(\boldsymbol{\theta})\sum_{s=n-k}^{n-1}F_i^{(s)}(\boldsymbol{\theta})$$

$$= h\sum_{k=1}^{n}\beta^k\sum_{s=n-k}^{n-1}\frac{1 - \beta^{s+1}}{1 - \beta}\sum_{i=1}^{d}\nabla_{ir}\mathcal{L}(\boldsymbol{\theta})\nabla_i\mathcal{L}(\boldsymbol{\theta})$$

$$= h \frac{\beta[1 - (2n+1)\beta^n(1-\beta) - \beta^{2n+1}]}{(1-\beta)^3} \sum_{i=1}^{d} \nabla_{ir}\mathcal{L}(\boldsymbol{\theta})\nabla_i\mathcal{L}(\boldsymbol{\theta}).$$

$\square$

**Lemma I.2** (Application of Corollary 3.3 to Example 1.3)**.** *Let $\{\boldsymbol{\theta}^{(n)}\}_{n\in\mathbb{Z}_{\geqslant 0}}$ be the sequence of vectors generated by the iteration in Equation (1) with initial condition $\boldsymbol{\theta}^{(0)}$, where $\boldsymbol{F}^{(n)}(\cdot)$ is as defined in Example 1.3, and the loss function $\mathcal{L}(\cdot)$ defined in an open convex bounded domain $\mathcal{D} \subset \mathbb{R}^d$ is three times continuously differentiable with bounded derivatives; also, let $T \geqslant 0$ be a fixed "time" horizon. Then Assumption 3.1 holds; in particular, by Corollary 3.3 the inequality $\max_{n\in[0:\lfloor T/h \rfloor]} \big\|\boldsymbol{\theta}^{(n)} - \tilde{\boldsymbol{\theta}}^{(n)}\big\|_\infty \leqslant C_2 h^2$ holds, where*

$$\tilde{\theta}_r^{(n+1)} = (1-\lambda h)\tilde{\theta}_r^{(n)} - h\left(\frac{\partial_r\mathcal{L}(\tilde{\boldsymbol{\theta}}^{(n)})}{\left(\left|\partial_r\mathcal{L}(\tilde{\boldsymbol{\theta}}^{(n)})\right|^2 + \varepsilon\right)^{1/2}} + M_r^{(n)}(\tilde{\boldsymbol{\theta}}^{(n)})\right),$$

$$M_r^{(n)}(\boldsymbol{\theta}) = h\left(\frac{\beta_1}{1-\beta_1} - \frac{(n+1)\beta_1^{n+1}}{1-\beta_1^{n+1}} - \frac{\beta_2}{1-\beta_2} + \frac{(n+1)\beta_2^{n+1}}{1-\beta_2^{n+1}} + \varepsilon\frac{\frac{\beta_2}{1-\beta_2} - \frac{(n+1)\beta_2^{n+1}}{1-\beta_2^{n+1}}}{|\partial_r\mathcal{L}(\boldsymbol{\theta})|^2 + \varepsilon}\right)$$
$$\times \frac{\left(\partial_r\|\nabla\mathcal{L}(\boldsymbol{\theta})\|_{1,\varepsilon} + \lambda[\nabla^2\mathcal{L}(\boldsymbol{\theta})\boldsymbol{\theta}]_r\right)}{(|\partial_r\mathcal{L}(\boldsymbol{\theta})|^2 + \varepsilon)^{1/2}}, \quad r \in [1:d].$$

*Proof.* The fact that Assumption 3.1 holds is already verified in Section C.

Next, in this case

$$F_r^{(n)}(\boldsymbol{\theta}) = \frac{\partial_r\mathcal{L}(\boldsymbol{\theta})}{\sqrt{\partial_r\mathcal{L}(\boldsymbol{\theta})^2 + \varepsilon}} + \lambda\theta_r,$$

$$\frac{\partial F_r^{(n)}}{\partial\theta_i^{(n-k)}}(\boldsymbol{\theta}) = \frac{\frac{1-\beta_1}{1-\beta_1^{n+1}}\beta_1^k\partial_{ir}\mathcal{L}(\boldsymbol{\theta})}{(|\partial_r\mathcal{L}(\boldsymbol{\theta})|^2 + \varepsilon)^{1/2}} - \frac{\frac{1-\beta_2}{1-\beta_2^{n+1}}\beta_2^k|\partial_r\mathcal{L}(\boldsymbol{\theta})|^2\partial_{ir}\mathcal{L}(\boldsymbol{\theta})}{(|\partial_r\mathcal{L}(\boldsymbol{\theta})|^2 + \varepsilon)^{3/2}}$$

Therefore,

$$M_r^{(n)}(\boldsymbol{\theta}) = h\sum_{k=1}^{n}\sum_{i=1}^{d}\frac{\partial F_r^{(n)}}{\partial\theta_i^{(n-k)}}(\boldsymbol{\theta})\sum_{s=n-k}^{n-1}F_i^{(s)}(\boldsymbol{\theta})$$

$$= h\sum_{i=1}^{d}\frac{\partial_{ir}\mathcal{L}(\boldsymbol{\theta})}{(|\partial_r\mathcal{L}(\boldsymbol{\theta})|^2 + \varepsilon)^{1/2}}\left[\frac{\partial_i\mathcal{L}(\boldsymbol{\theta})}{(|\partial_i\mathcal{L}(\boldsymbol{\theta})|^2 + \varepsilon)^{1/2}} + \lambda\theta_i\right]\frac{1-\beta_1}{1-\beta_1^{n+1}}\sum_{k=1}^{n}k\beta_1^k$$

$$- h|\partial_r\mathcal{L}(\boldsymbol{\theta})|^2\sum_{i=1}^{d}\frac{\partial_{ir}\mathcal{L}(\boldsymbol{\theta})}{(|\partial_r\mathcal{L}(\boldsymbol{\theta})|^2 + \varepsilon)^{3/2}}\left[\frac{\partial_i\mathcal{L}(\boldsymbol{\theta})}{(|\partial_i\mathcal{L}(\boldsymbol{\theta})|^2 + \varepsilon)^{1/2}} + \lambda\theta_i\right]\frac{1-\beta_2}{1-\beta_2^{n+1}}\sum_{k=1}^{n}k\beta_2^k$$

$$= h\left(\frac{\beta_1}{1-\beta_1} - \frac{(n+1)\beta_1^{n+1}}{1-\beta_1^{n+1}}\right)\sum_{i=1}^{d}\frac{\partial_{ir}\mathcal{L}(\boldsymbol{\theta})}{(|\partial_r\mathcal{L}(\boldsymbol{\theta})|^2 + \varepsilon)^{1/2}}\left[\frac{\partial_i\mathcal{L}(\boldsymbol{\theta})}{(|\partial_i\mathcal{L}(\boldsymbol{\theta})|^2 + \varepsilon)^{1/2}} + \lambda\theta_i\right]$$

$$- h\left(\frac{\beta_2}{1-\beta_2} - \frac{(n+1)\beta_2^{n+1}}{1-\beta_2^{n+1}}\right)\sum_{i=1}^{d}\frac{|\partial_r\mathcal{L}(\boldsymbol{\theta})|^2\partial_{ir}\mathcal{L}(\boldsymbol{\theta})}{(|\partial_r\mathcal{L}(\boldsymbol{\theta})|^2 + \varepsilon)^{3/2}}\left[\frac{\partial_i\mathcal{L}(\boldsymbol{\theta})}{(|\partial_i\mathcal{L}(\boldsymbol{\theta})|^2 + \varepsilon)^{1/2}} + \lambda\theta_i\right]$$

$$= h\left(\frac{\beta_1}{1-\beta_1} - \frac{(n+1)\beta_1^{n+1}}{1-\beta_1^{n+1}}\right)\frac{\left(\partial_r\|\nabla\mathcal{L}(\boldsymbol{\theta})\|_{1,\varepsilon} + \lambda[\nabla^2\mathcal{L}(\boldsymbol{\theta})\boldsymbol{\theta}]_r\right)}{(|\partial_r\mathcal{L}(\boldsymbol{\theta})|^2 + \varepsilon)^{1/2}}$$

$$- h\left(\frac{\beta_2}{1-\beta_2} - \frac{(n+1)\beta_2^{n+1}}{1-\beta_2^{n+1}}\right)\frac{|\partial_r\mathcal{L}(\boldsymbol{\theta})|^2\left(\partial_r\|\nabla\mathcal{L}(\boldsymbol{\theta})\|_{1,\varepsilon} + \lambda[\nabla^2\mathcal{L}(\boldsymbol{\theta})\boldsymbol{\theta}]_r\right)}{(|\partial_r\mathcal{L}(\boldsymbol{\theta})|^2 + \varepsilon)^{3/2}}$$

$$= h\left(\frac{\beta_1}{1-\beta_1} - \frac{(n+1)\beta_1^{n+1}}{1-\beta_1^{n+1}} - \frac{\beta_2}{1-\beta_2} + \frac{(n+1)\beta_2^{n+1}}{1-\beta_2^{n+1}} + \varepsilon\frac{\frac{\beta_2}{1-\beta_2} - \frac{(n+1)\beta_2^{n+1}}{1-\beta_2^{n+1}}}{|\partial_r\mathcal{L}(\boldsymbol{\theta})|^2 + \varepsilon}\right)$$

$$\times \frac{\left(\partial_r \|\nabla \mathcal{L}(\boldsymbol{\theta})\|_{1,\varepsilon} + \lambda [\nabla^2 \mathcal{L}(\boldsymbol{\theta})\boldsymbol{\theta}]_r\right)}{(|\partial_r \mathcal{L}(\boldsymbol{\theta})|^2 + \varepsilon)^{1/2}}$$

as desired. $\qquad\square$

**Lemma I.3** (Application of Corollary 3.3 to Example 1.5). *Let* $\{\boldsymbol{\theta}^{(n)}\}_{n \in \mathbb{Z}_{\geqslant 0}}$ *be the sequence of vectors generated by the iteration in Equation* (1) *with initial condition* $\boldsymbol{\theta}^{(0)}$, *where* $\boldsymbol{F}^{(n)}(\cdot)$ *is as defined in Example* 1.5, *the function* $\mathcal{K}(\cdot) : \mathcal{M} \to \mathbb{R}$ *defined in a open bounded region* $\mathcal{M} \subset \mathbb{R}^d$ *is three times continuously differentiable with bounded derivatives, and the loss function* $\mathcal{L}(\cdot)$ *defined in an open convex bounded domain* $\mathcal{D} \subset \mathbb{R}^d$ *is three times continuously differentiable with bounded derivatives; also, let* $T \geqslant 0$ *be a fixed "time" horizon. Then Assumption* 3.1 *holds; in particular, by Corollary* 3.3 *the inequality* $\max_{n \in [0:\lfloor T/h \rfloor]} \|\boldsymbol{\theta}^{(n)} - \tilde{\boldsymbol{\theta}}^{(n)}\|_\infty \leqslant C_2 h^2$ *holds, where*

$$\tilde{\theta}_r^{(n+1)} = (1 - \lambda h)\tilde{\theta}_r^{(n)} - h\left(-\partial_r \mathcal{K}\left(-(1 - \rho_1 \rho_2^n)\nabla \mathcal{L}(\tilde{\boldsymbol{\theta}}^{(n)})\right) + M_r^{(n)}(\tilde{\boldsymbol{\theta}}^{(n)})\right),$$

$$M_r^{(n)}(\boldsymbol{\theta}) = h\rho_1(1 - \rho_2)\sum_{i=1}^{d}\sum_{j=1}^{d} \partial_{jr}\mathcal{K}\left(-(1 - \rho_1 \rho_2^n)\nabla \mathcal{L}(\boldsymbol{\theta})\right)\partial_{ij}\mathcal{L}(\boldsymbol{\theta})$$

$$\times \sum_{k=1}^{n} \rho_2^{k-1} \sum_{s=n-k}^{n-1} \left(-\partial_i \mathcal{K}\left(-(1 - \rho_1 \rho_2^s)\nabla \mathcal{L}(\boldsymbol{\theta})\right) + \lambda\theta_i\right)$$

$$= h\frac{\rho_1}{1 - \rho_2}\sum_{i=1}^{d}\sum_{j=1}^{d} \partial_{jr}\mathcal{K}\left(-\nabla \mathcal{L}(\boldsymbol{\theta})\right)\partial_{ij}\mathcal{L}(\boldsymbol{\theta})\left(-\partial_i \mathcal{K}\left(-\nabla \mathcal{L}(\boldsymbol{\theta})\right) + \lambda\theta_i\right) + o_n(1),$$

*where* $o_n(1)$ *is a function of* $\boldsymbol{\theta}$ *converging to zero uniformly in* $\boldsymbol{\theta} \in \mathcal{D}$.

*Proof.* The fact that Assumption 3.1 holds is already verified in Section C.

Next, in this case

$$F_r^{(n)}(\boldsymbol{\theta}) = -\partial_r \mathcal{K}\left(-(1 - \rho_1 \rho_2^n)\nabla \mathcal{L}(\boldsymbol{\theta})\right) + \lambda\theta_r,$$

$$\frac{\partial F_r^{(n)}}{\partial \theta_i^{(n-k)}}(\boldsymbol{\theta}) = \sum_{j=1}^{d} \partial_{jr}\mathcal{K}\left(-(1 - \rho_1 \rho_2^n)\nabla \mathcal{L}(\boldsymbol{\theta})\right)(1 - \rho_2)\rho_1 \rho_2^{k-1}\partial_{ij}\mathcal{L}(\boldsymbol{\theta}).$$

Therefore,

$$M_r^{(n)}(\boldsymbol{\theta}) = h\sum_{k=1}^{n}\sum_{i=1}^{d} \frac{\partial F_r^{(n)}}{\partial \theta_i^{(n-k)}}(\boldsymbol{\theta}) \sum_{s=n-k}^{n-1} F_i^{(s)}(\boldsymbol{\theta})$$

$$= h\rho_1(1 - \rho_2)\sum_{i=1}^{d}\sum_{j=1}^{d} \partial_{jr}\mathcal{K}\left(-(1 - \rho_1 \rho_2^n)\nabla \mathcal{L}(\boldsymbol{\theta})\right)\partial_{ij}\mathcal{L}(\boldsymbol{\theta})$$

$$\times \sum_{k=1}^{n} \rho_2^{k-1} \sum_{s=n-k}^{n-1} \left(-\partial_i \mathcal{K}\left(-(1 - \rho_1 \rho_2^s)\nabla \mathcal{L}(\boldsymbol{\theta})\right) + \lambda\theta_i\right)$$

Note that

$$\partial_i \mathcal{K}(-\nabla \mathcal{L}(\boldsymbol{\theta})) - \partial_i \mathcal{K}\left(-(1 - \rho_1 \rho_2^s)\nabla \mathcal{L}(\boldsymbol{\theta})\right) = -\left[\nabla^2 \mathcal{K}(\boldsymbol{\zeta})\nabla \mathcal{L}(\boldsymbol{\theta})\right]_i \rho_1 \rho_2^s,$$

where $\boldsymbol{\zeta}$ is on the segment between $-(1 - \rho_1 \rho_2^s)\nabla \mathcal{L}(\boldsymbol{\theta})$ and $-\nabla \mathcal{L}(\boldsymbol{\theta})$. Applying Lemma H.2 with $a_k = \rho_2^{k-1}$ and $b_s = \rho_1 \rho_2^s$ we see that

$$\sum_{k=1}^{n} \rho_2^{k-1} \sum_{s=n-k}^{n-1} \left(-\partial_i \mathcal{K}\left(-(1 - \rho_1 \rho_2^s)\nabla \mathcal{L}(\boldsymbol{\theta})\right) + \lambda\theta_i\right)$$

$$= \sum_{k=1}^{n} \rho_2^{k-1} k\left(-\partial_i \mathcal{K}\left(-\nabla \mathcal{L}(\boldsymbol{\theta})\right) + \lambda\theta_i\right) + o_n(1) = \frac{1}{(1 - \rho_2)^2}\left(-\partial_i \mathcal{K}\left(-\nabla \mathcal{L}(\boldsymbol{\theta})\right) + \lambda\theta_i\right) + o_n(1),$$

where $o_n(1) \to 0$ as $n \to \infty$ uniformly in $\boldsymbol{\theta} \in \mathcal{D}$. Using the boundedness of derivatives again, we also have

$$h\rho_1(1 - \rho_2) \sum_{i=1}^{d} \sum_{j=1}^{d} \partial_{jr}\mathcal{K}\big(-(1 - \rho_1\rho_2^n)\nabla\mathcal{L}(\boldsymbol{\theta})\big)\partial_{ij}\mathcal{L}(\boldsymbol{\theta})$$

$$= h\rho_1(1 - \rho_2) \sum_{i=1}^{d} \sum_{j=1}^{d} \partial_{jr}\mathcal{K}\big(-\nabla\mathcal{L}(\boldsymbol{\theta})\big)\partial_{ij}\mathcal{L}(\boldsymbol{\theta}) + o_n(1)$$

and

$$M_r^{(n)}(\boldsymbol{\theta}) = h\frac{\rho_1}{1 - \rho_2} \sum_{i=1}^{d} \sum_{j=1}^{d} \partial_{jr}\mathcal{K}\big(-\nabla\mathcal{L}(\boldsymbol{\theta})\big)\partial_{ij}\mathcal{L}(\boldsymbol{\theta})\big(-\partial_i\mathcal{K}\big(-\nabla\mathcal{L}(\boldsymbol{\theta})\big) + \lambda\theta_i\big) + o_n(1)$$

as desired. $\qquad\qquad\square$

## J   Empirical Illustrations

As a sanity check, we verify that the memoryless iteration (4) that we find is closer than the first-order approximation (the one with no correction from memory: for example, the first-order approximation of Adam is sign gradient descent [47]). We see in Figure 1 that the $\ell_\infty$ (maximal) norm of the weight difference is smaller with the correction term. Note that the learning rates are realistic and weight decay is present.

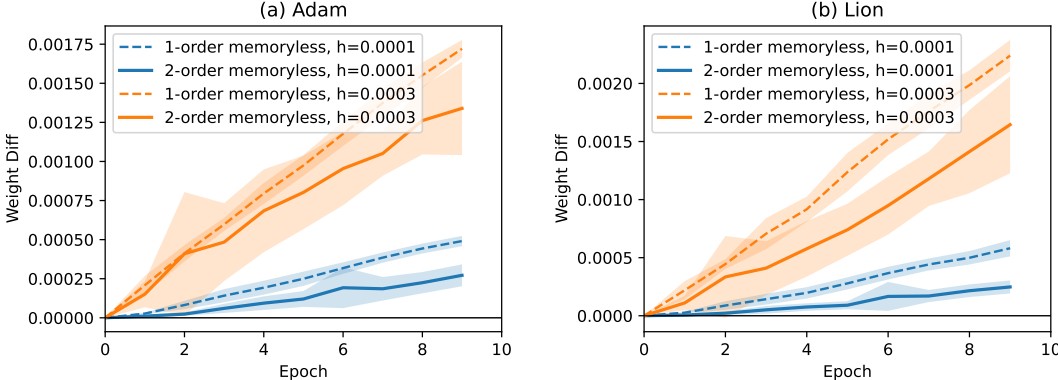

Figure 1: Comparison of the trajectories: $\ell_\infty$-norm of weight difference between the second-order memoryless method from Theorem 3.2 and the first-order approximation (signGD): **(a)** Adam, **(b)** Lion (perturbed by $\varepsilon$ and with bias correction). MLP with GELU activation on MNIST-10K, learning rates $h \in \{10^{-4}, 3 \times 10^{-4}\}$, weight decay $10^{-3}/h$, $\varepsilon = 10^{-6}$.

As a preliminary empirical illustration, we train ResNet-50 [31] on CIFAR-10 [38] using Adam (with decoupled weight decay) and Lion. We keep the $\beta_1$ parameter of Adam at $0.99$ (for stable training on CIFAR-10 [47, 11]) and sweep the $\beta_2$ parameter. We plot in Figure 2(a) the test accuracy at a fixed small training loss threshold (controlling for training speed). As predicted in Section 4 and confirming the observation from [11] for pure Adam without weight decay, the test accuracy drops as $\beta_2$ approaches one. We see that Adam with lower values of $\beta_2$ can sometimes outperform Lion with default hyperparameters and thus close the generalization gap between these two algorithms, consistent with the theory. We also observe this phenomenon on a language task by training Transformer-XL [18] on WikiText-2 [49]. We fix the default $\beta_1 = 0.9$ for Adam and sweep $\beta_2$, plotting the minimal validation perplexity achieved before overfitting; as in the vision task, we compare with Lion whose hyperparameters are set at default values. We observe in Figure 2(b) the same trends as above (in large-batch training, higher $\beta_2$ increases the best validation perplexity, that is, hurts generalization; sometimes, taking lower $\beta_2$ can close the gap between Adam and Lion).

We provide some additional sweeps with different learning rates in Figures 3 and 4.

The code is available at `https://github.com/borshigida/how-memory-modifies-loss`.

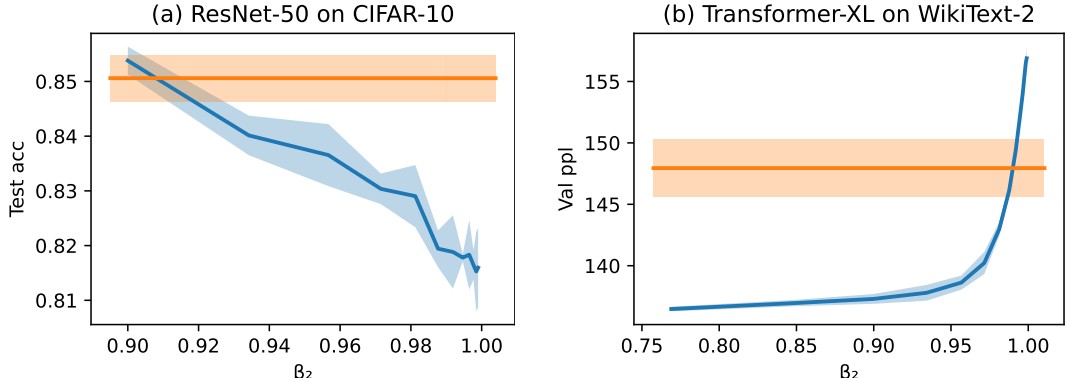

Figure 2: **(a)** ResNet-50 on CIFAR-10: test accuracy at training loss threshold 0.05. Full-batch Adam, learning rate $h = 10^{-3.5}$, $\beta_1 = 0.99$, $\varepsilon = 10^{-6}$, weight decay 0.005. For comparison, we also show Lion with the same learning rate and weight decay (with default $\rho_1 = 0.9$, $\rho_2 = 0.99$). **(b)** Minimal validation perplexity (before overfitting) of Transformer-XL trained with full-batch Adam on WikiText-2 with learning rate $10^{-4}$, $\beta_1 = 0.9$, $\varepsilon = 10^{-6}$. For comparison, we also show Lion (with default $\rho_1 = 0.9$, $\rho_2 = 0.99$). All results are averaged over three iterations.

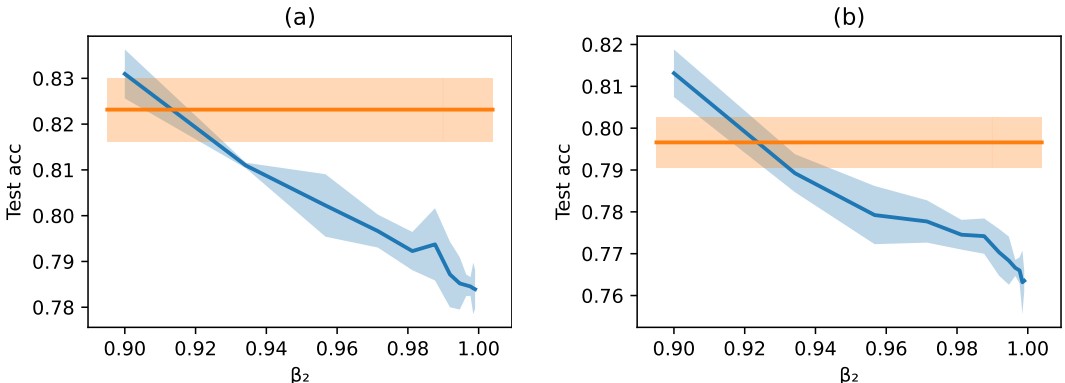

Figure 3: ResNet-50 on CIFAR-10: test accuracy at training loss threshold 0.05. Full-batch Adam, learning rates **(a)** $h = 10^{-4}$; **(b)** $h = 5 \times 10^{-5}$, $\beta_1 = 0.99$, $\varepsilon = 10^{-6}$, weight decay $5 \times 10^{-6.5}/h$. For comparison, we also show Lion with the same learning rates and weight decay (with default $\rho_1 = 0.9$, $\rho_2 = 0.99$). All results are averaged over three iterations.

## J.1   A Note on the Edge of Stability and Comparisons with Lion on Vision Tasks

Cohen et al. [16] notice that in a sense Adam trains at the edge of stability. They view Adam as momentum gradient descent with evolving preconditioner

$$\mathbf{P}_{t+1} = (1 - \beta_1^{t+1})\left[\text{diag}\left(\sqrt{\frac{\boldsymbol{\nu}_{t+1}}{1 - \beta_2^{t+1}}}\right) + \epsilon\mathbf{I}\right].$$

They define "preconditioned sharpness" to be the top eigenvalue of the preconditioned Hessian $\lambda_1(\mathbf{P}_t^{-1}\mathbf{H}_t)$, where $\mathbf{H}_t$ is the Hessian of the loss, and observe that this quantity often oscillates around the stability threshold $\frac{2+2\beta_1}{(1-\beta_1)\eta}$, where $\eta$ is the learning rate. (This fraction comes from the fact that if the preconditioner were constant, Adam would become a form of preconditioned gradient descent with EMA-style momentum, and this is the ordinary stability threshold of EMA-style heavy-ball momentum on the quadratic Taylor approximation of the loss; we refer to Cohen et al. [16] for details.) They use large-batch training on CIFAR-10/100. We train a CNN on CIFAR-10 as well, and reproduce this result in Figure 5. We also plot ordinary sharpness $\lambda_1(\mathbf{H}_t)$ (top hessian eigenvalue), which first increases and then decreases. Recall that this is an extremely unstable regime of training [47].

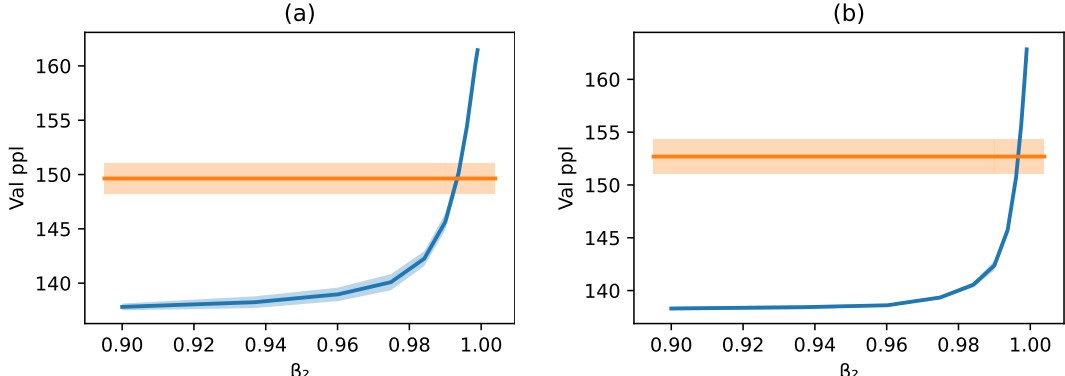

Figure 4: Minimal validation perplexity (before overfitting) of Transformer-XL trained with full-batch Adam on WikiText-2 with learning rates **(a)** $h = 5 \times 10^{-5}$; **(b)** $h = 2.5 \times 10^{-5}$, weight decay $10^{-8}/h$, $\beta_1 = 0.9$, $\varepsilon = 10^{-6}$. For comparison, we also show Lion with the same learning rates and weight decay (with default $\rho_1 = 0.9$, $\rho_2 = 0.99$). All results are averaged over three iterations.

Note that the parameter controlling the exponential forgetting of gradients $\beta_1$ corresponds to the $\rho_2$ parameter of Lion, so the default $\rho_2 = 0.99$ in Lion would match $\beta_1 = 0.99$ rather than $\beta_1 = 0.9$ in Adam. If we take $\beta_1 = 0.99$ which is the "smooth" regime of training, preconditioned sharpness does not reach the stability threshold (Figure 6). Note also that ordinary sharpness $\lambda_1(\mathbf{H}_t)$ (top hessian eigenvalue) is much lower for small $\beta_2$ (especially noticeable for $\beta_1 \leqslant 0.9$). This suggests that in large-batch training on vision tasks, taking $\beta_2 < \beta_1 = 0.99$ strongly regularizes training, moving the model parameters to flatter regions of the loss space. It is a promising direction to investigate the limits of such regularization: for example, taking $\beta_2$ near the threshold of divergence may regularize training so much that the default Lion will not be able to match in terms of generalization error.

## K  Experiment Details and Licenses

Our implementation of ResNet-50 follows the one from [11] (small modification of the standard `torchvision` implementation to allow for training on CIFAR-10 rather than ImageNet). The `torchvision` repository has the BSD 3-Clause license. CIFAR-10 is released without an explicit license. MNIST has the CC BY-SA 3.0 license.

Our implementation of training Transformer-XL on WikiText-2 follows the one from [41] which is a small modification of the codebase[2] for [18], licensed under the Apache-2.0 License. The WikiText-2 dataset is released under the CC BY-SA 3.0 license.

For Adam (with decoupled weight decay), we use the standard implementation from `pytorch.optim`; Lion is taken from the `google/automl` repository[3]. This repository is licensed under the Apache License 2.0. The implementations of the optimizers used for comparing the trajectories in Figure 1 of the paper are custom and match exactly our analytical formulas (in particular, Lion has bias correction and the soft-sign function $x \mapsto x/\sqrt{x^2 + \varepsilon}$ instead of the sign function), given below.

**AdamW: memoryless update**   The (full-batch) memoryless AdamW approximation is

$$\theta_j^{(n+1)} = \theta_j^{(n)} - hF_j^{(n)}(\boldsymbol{\theta}^{(n)}) - hM_j^{(n)}(\boldsymbol{\theta}^{(n)}),$$

where

$$F_j^{(n)}(\boldsymbol{\theta}) = \frac{\nabla_j \mathcal{L}(\boldsymbol{\theta})}{\sqrt{|\nabla_j \mathcal{L}(\boldsymbol{\theta})|^2 + \varepsilon}} + \lambda \theta_j,$$

$$M_j^{(n)}(\boldsymbol{\theta}) = -h\left(\frac{\beta_2}{1-\beta_2} - \frac{(n+1)\beta_2^{n+1}}{1-\beta_2^{n+1}}\right)\frac{|\nabla_j \mathcal{L}(\boldsymbol{\theta})|^2\big(\nabla_j\|\nabla \mathcal{L}(\boldsymbol{\theta})\|_{1,\varepsilon} + \lambda[\nabla^2\mathcal{L}(\boldsymbol{\theta})\boldsymbol{\theta}]_j\big)}{(|\nabla_j \mathcal{L}(\boldsymbol{\theta})|^2 + \varepsilon)^{3/2}}$$

---

[2]https://github.com/kimiyoung/transformer-xl
[3]https://github.com/google/automl/tree/master/lion

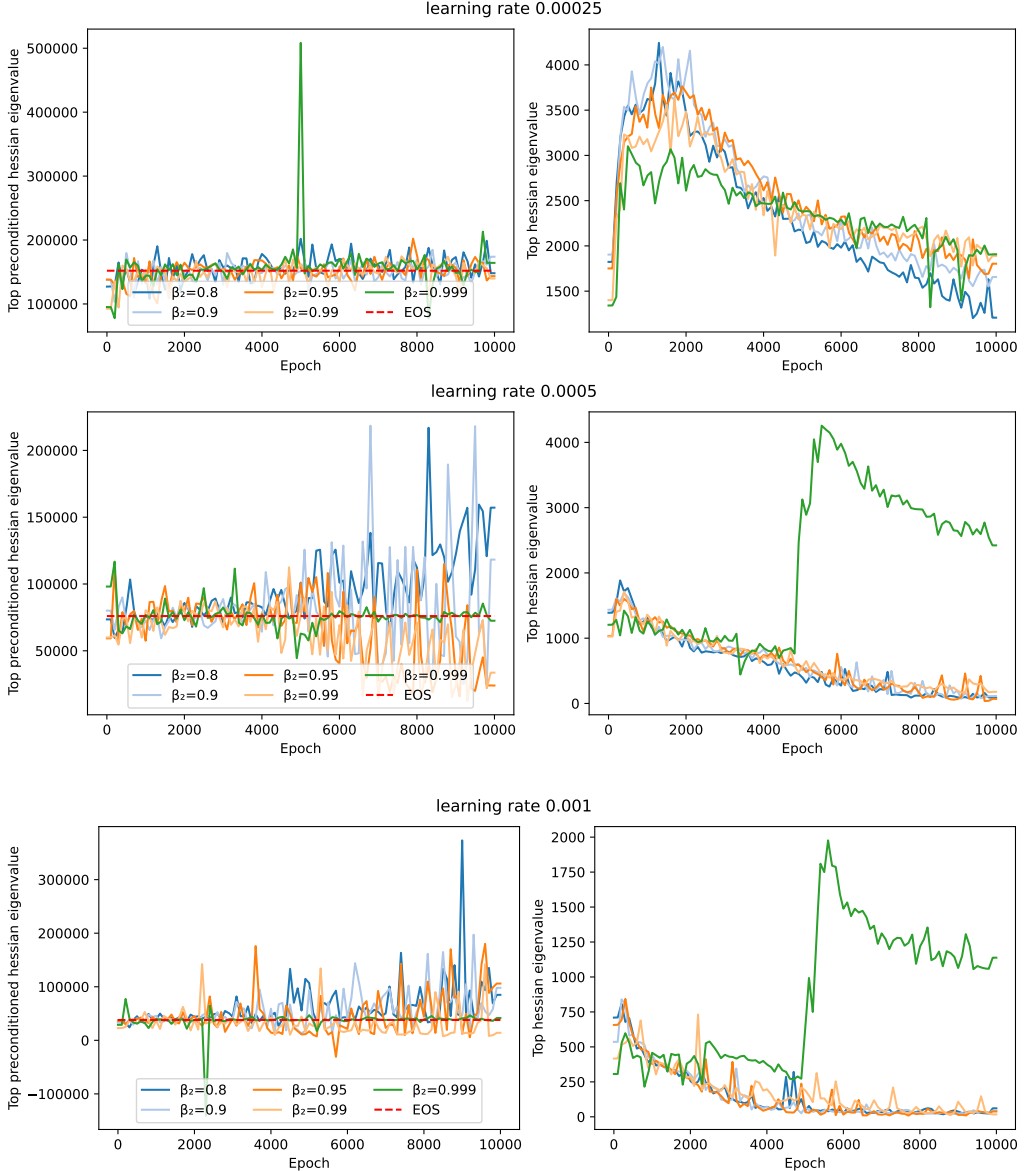

Figure 5: CNN trained on CIFAR-10 with full-batch Adam, $\beta_1 = 0.9$, $\epsilon = 10^{-6}$, learning rate $0.001$. Left: preconditioned sharpness $\lambda_1(\mathbf{P}_t^{-1}\mathbf{H}_t)$ oscillates around the stability threshold. Right: the plots of ordinary sharpness $\lambda_1(\mathbf{H}_t)$.

$$+ h\left(\frac{\beta_1}{1-\beta_1} - \frac{(n+1)\beta_1^{n+1}}{1-\beta_1^{n+1}}\right) \frac{\left(\nabla_j \|\nabla\mathcal{L}(\boldsymbol{\theta})\|_{1,\varepsilon} + \lambda[\nabla^2\mathcal{L}(\boldsymbol{\theta})\boldsymbol{\theta}]_j\right)}{\sqrt{|\nabla_j\mathcal{L}(\boldsymbol{\theta})|^2 + \varepsilon}}.$$

**Lion-$\mathcal{K}$ with bias correction**  The Lion-$\mathcal{K}$ algorithm with bias correction is defined as in Example 1.5 of the paper except bias correction is added:

$$\boldsymbol{F}^{(n)}(\boldsymbol{\theta}^{(n)}, \dots, \boldsymbol{\theta}^{(0)}) = -\nabla\mathcal{K}(\boldsymbol{m}_1^{(n+1)} + \boldsymbol{m}_2^{(n+1)}) + \boldsymbol{m}_3^{(n+1)},$$

$$\text{where} \quad \boldsymbol{m}_1^{(n+1)} = -\frac{1-\rho_2}{1-\rho_2^{n+1}}\frac{\rho_1}{\rho_2}\sum_{k=0}^{n}\rho_2^{n-k}\nabla\mathcal{L}(\boldsymbol{\theta}^{(k)}),$$

$$\boldsymbol{m}_2^{(n+1)} = -\left(1 - \frac{\rho_1}{\rho_2}\right)\nabla\mathcal{L}(\boldsymbol{\theta}^{(n)}),$$

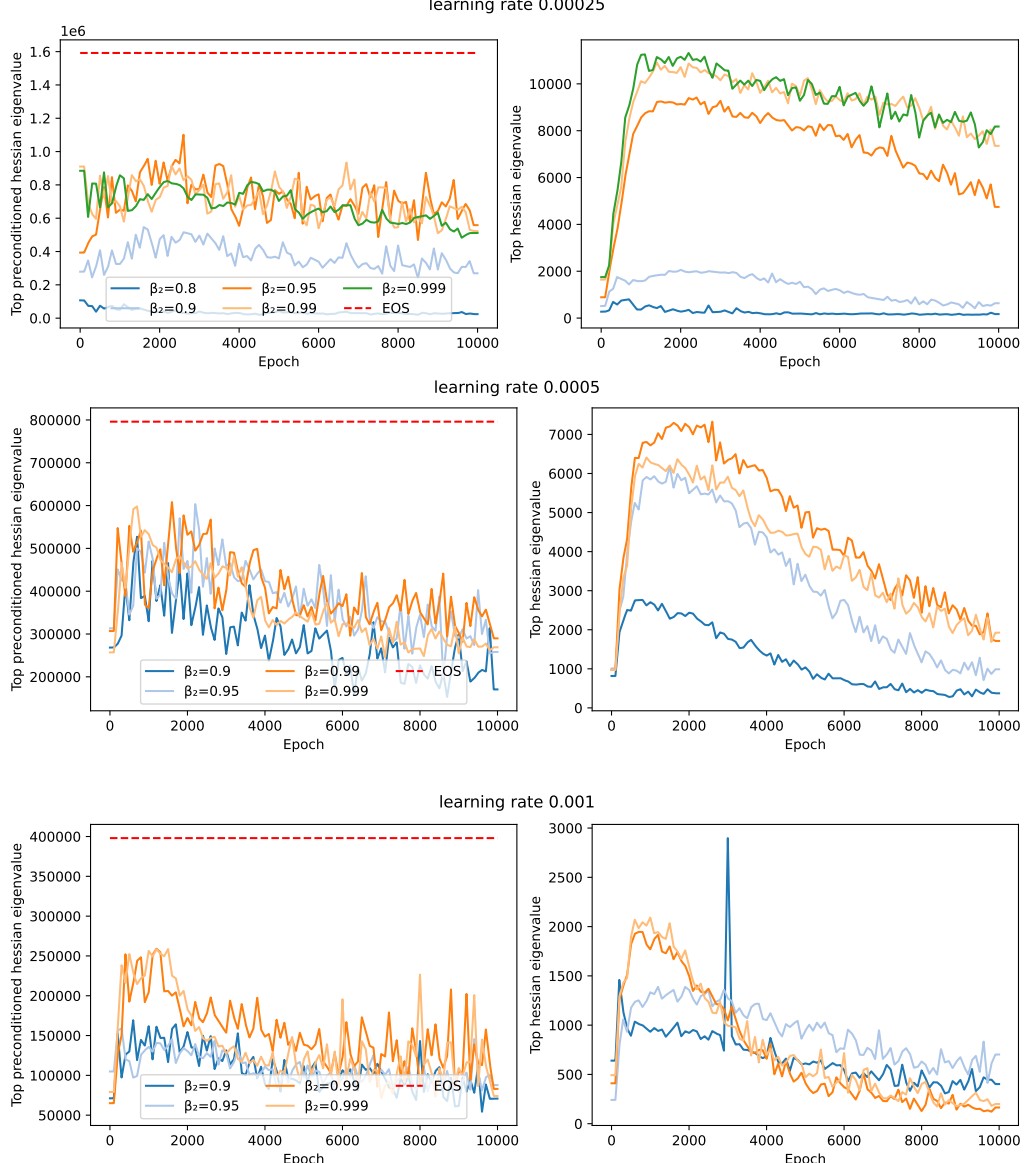

Figure 6: CNN trained on CIFAR-10 with full-batch Adam, $\beta_1 = 0.99$, $\epsilon = 10^{-6}$, learning rate 0.001. Left: preconditioned sharpness $\lambda_1(\mathbf{P}_t^{-1}\mathbf{H}_t)$ does not reach the stability threshold. Right: the plots of ordinary sharpness $\lambda_1(\mathbf{H}_t)$.

$$\boldsymbol{m}_3^{(n+1)} = \lambda\boldsymbol{\theta}^{(n)}.$$

**Lion (perturbed by $\varepsilon$): memoryless update**    The memoryless iteration is given by

$$\theta_j^{(n+1)} = \theta_j^{(n)} - hF_j^{(n)}(\boldsymbol{\theta}^{(n)}) - hM_j^{(n)}(\boldsymbol{\theta}^{(n)}),$$

where

$$F_j^{(n)}(\boldsymbol{\theta}) = \frac{\nabla_j\mathcal{L}(\boldsymbol{\theta})}{\sqrt{|\nabla_j\mathcal{L}(\boldsymbol{\theta})|^2 + \varepsilon}} + \lambda\theta_j,$$

$$M_j^{(n)}(\boldsymbol{\theta}) = h\left[\frac{\rho_1}{1-\rho_2} - \frac{(n+1)\rho_2^n\rho_1}{1-\rho_2^{n+1}}\right]$$

$$\times \frac{\varepsilon}{\left(|\nabla_j \mathcal{L}(\boldsymbol{\theta})|^2 + \varepsilon\right)^{3/2}} \nabla_j \left[\|\nabla \mathcal{L}(\boldsymbol{\theta})\|_{1,\varepsilon} + \lambda(\nabla \mathcal{L}(\boldsymbol{\theta})^\mathsf{T}\boldsymbol{\theta} - \mathcal{L}(\boldsymbol{\theta}))\right].$$

### K.1 Compute Resources

One sweep of hyperparameter $\beta_2$ contained about 12 runs, with each run repeated for 3 iterations. Each run took about 10 hours on average on one machine with a devoted 40 GB NVIDIA A100 GPU (though the training horizon was longer than necessary). This puts compute resources at around $12 \times 10 \times 3 = 360$ A100-GPU-hours per sweep. In Figure 2, two sweeps were conducted. The experiments on truncated MNIST conducted to produce Figure 1 used negligible resources compared to the sweeps described (less than 1 GPU-hour). Additional compute was used for preliminary experimentation.

