# OpenReview forum: "How Memory in Optimization Algorithms Implicitly Modifies the Loss"
_NeurIPS.cc/2025/Conference — NeurIPS 2025 poster_

### Official Review · Reviewer_CQS8 · 2025-06-27

**Clarity:** 4
**Significance:** 3
**Originality:** 3
**Rating:** 5
**Confidence:** 4

**Summary:**

This paper introduces a general framework to study how the memory (which is usually the momentum term) affects the implicit bias. Specifically, for each algorithm, such as Adam and Lion, they propose a memoryless version by introducing a correction term. This correction term can be associated with a regularizer, which corresponds to the source of implicit bias. As an application, they show Lion does not have the kind of implicit anti-regularization induced by memory that AdamW does, which might be the reason why Lion performs better than Adam in practice.

**Questions:**

- In Lion-K's original paper, as well as shown in other papers, algorithms such Adam and Lion themselves have an implicit regularization. In this paper, they show that the memory has an additional implicit bias. Is it possible to combine them to derive the total implicit regularization effect?

- Can you provide a simulation to see whether Lion penalizes more than Adam on the L1-norm?

**Ethical Concerns:**

["NO or VERY MINOR ethics concerns only"]

**Final Justification:**

I maintain my original score.

**Limitations:**

Yes

**Quality:**

3

**Strengths And Weaknesses:**

**Strengths:**
- The paper is well-written. The derivation of the memoryless variant and the way it relates to an implicit regularizer is well-presented.
- The results on Lion and Adam are new and interesting.

**Weaknesses:**
- Probably more examples, such as Muon could be added.

---

> ### Author Rebuttal · Authors · 2025-07-29
>
> We greatly appreciate your assessment and feedback. Your questions are addressed below.
>
> **Q1.** This is a very important point which we thank you for raising. By the original Lion-$\mathcal{K}$ paper and the related work [56], full-batch AdamW and Lion, just like their first order approximations (sign descent), converge to a KKT point of the constrained minimization of the loss subject to $\\| \boldsymbol{\theta} \\|\_{\infty} \leq 1 / \lambda$. Our analysis suggests that AdamW is approximately preconditioned gradient descent whose loss is rescaled and perturbed by the function $\\| \nabla \mathcal{L}(\boldsymbol{\theta}) \\|\_1 + \lambda \nabla \mathcal{L}(\boldsymbol{\theta})^T \boldsymbol{\theta}$. The KKT points of the optimization problem are precisely those that satisfy the constraint and minimize this function, that is, $\\| \boldsymbol{\theta}^{\star} \\|\_{\infty} \leq 1 / \lambda$ and $\\| \nabla \mathcal{L}(\boldsymbol{\theta}^{\star}) \\|\_1 + \lambda \nabla \mathcal{L}(\boldsymbol{\theta}^{\star})^T \boldsymbol{\theta}^{\star} = \min\_{\\| \boldsymbol{\theta} \\|\_{\infty} \leq 1 / \lambda} [ \\| \nabla \mathcal{L}(\boldsymbol{\theta}) \\|\_1 + \lambda \nabla \mathcal{L}(\boldsymbol{\theta})^T \boldsymbol{\theta}] = 0$. Thus, these two frameworks are distinct but consistent with each other: one deals with the implicit bias of weight decay in versions of sign descent, and the other identifies the effect of memory. Intuitively, memory does not change the optimization problem but changes the way the algorithm solves it (and near the solution the loss perturbation is at a minimum).
>
> **Q2.** We have plotted how the one-norm behaves as a function of the epoch during the training run as in Figure 2(a), and will include the figure into the revised version of the work. (We apply exponential moving averaging (`pandas.DataFrame.ewm(alpha=0.1)`) to smooth out the curves.) Since we are forbidden from including pictures, we resort to a description here. For AdamW, the one-norm decreases rapidly during the initial ~120 epochs, after which it starts increasing again, peaks at around epoch 350-400, and further decreases. The peak values increase as $\beta_2$ increases:
>
> ```
> adam β₂=0.9: peak one norm=618.1 at epoch=389
> adam β₂=0.9342066775342432: peak one norm=802.9 at epoch=364
> adam β₂=0.9567123871891694: peak one norm=1017 at epoch=333
> adam β₂=0.971519641315642: peak one norm=985.8 at epoch=355
> adam β₂=0.9812618257713962: peak one norm=1188 at epoch=363
> adam β₂=0.9876715326055794: peak one norm=1190 at epoch=348
> adam β₂=0.9918886916921031: peak one norm=1174 at epoch=370
> adam β₂=0.9946633007687937: peak one norm=1231 at epoch=352
> adam β₂=0.9964888082657849: peak one norm=1214 at epoch=348
> adam β₂=0.9976898702999168: peak one norm=1250 at epoch=333
> adam β₂=0.9984800889170471: peak one norm=1277 at epoch=352
> adam β₂=0.999: peak one norm=1324 at epoch=358
> ```
>
> The one-norm of the network parameters trained with Lion does not contain such pronounced hills but rather has rapid significant jumps. Apart from jumps, the Lion one-norm curve stays below the Adam curves, reaching value ~650 by roughly epoch 300.
>
> Please let us know if you would like to discuss any additional points or you have further questions or comments. Thank you!

---

> > ### Comment · Reviewer_CQS8 · 2025-08-05
> >
> > Thanks for your response and for sharing the experiment. I don’t have any further questions about the paper. Just a quick thought on the observations from your experiment: there might be a way to explicitly control the $\ell_1$-norm in both Adam and Lion, which could potentially improve training stability and enhance model performance.

---

> > > ### Author Response · Authors · 2025-08-07
> > >
> > > Thank you for your additional thought regarding the potential advantages of penalizing the $\ell_1$-norm explicitly. We will explore this idea in future research.

---

### Official Review · Reviewer_Fq9j · 2025-06-30

**Clarity:** 3
**Significance:** 3
**Originality:** 3
**Rating:** 5
**Confidence:** 4

**Summary:**

The paper presents a unified theoretical framework that connects various memory-based optimizers. Through further analysis, the authors demonstrate how a memory-based optimizer can be reformulated as an equivalent memory-less update augmented with an explicit correction term.
By collapsing the historical dependence $F^{(n)}(\theta^{(n)}, \ldots, \theta^{(0)})$
Onto $F^{(n)}(\theta^{(n)}, \ldots, \theta^{(n)})$  and adding a first-order Taylor remainder
$M^{(n)}(\theta)$, the authors:

* Show that momentum in gradient descent (GD) implicitly adds the penalty
    $\frac{h \beta^2}{(1 - \beta)^2} \| \nabla L(\theta) \|^2$, explaining its “flatter-minima” bias.
* Derive a rigorous one-step and an $\mathcal{O}(h^2)$ global error bound between the real and memory-less trajectories.
* Apply the framework to AdamW and Lion. They prove that AdamW’s two-moment memory contributes an anti-regularization term, whereas Lion’s sign-based memory does not, offering a theoretical reason Lion usually generalizes better.
* Provide empirical checks (MNIST, CIFAR-10, WikiText-2) showing that the memory-less approximations track the real optimizers more closely than first-order baselines, and that the predicted generalization trends indeed hold.

**Questions:**

* Your bound yields an $O(h^{2})$ global error, yet typical Transformer training uses  $h \approx 10^{-4}\!-\!10^{-3}$.   Could you plot trajectory error vs. learning rate to illustrate where the approximation breaks down?
* Optimizers such as AdaGrad or Shampoo employ a time–varying matrix pre-conditioner.  Do the fast–decay and Taylor arguments still hold when $F^{(n)}$ contains a changing matrix $P^{(n)}$?  A brief discussion (or lemma) would broaden the framework’s scope.
* Appendix~G derives an extra noise-regularization term, yet the main experiments are full-batch.  A small-batch comparison of AdamW and Lion (e.g.\ CIFAR-10 with batch 128) would directly verify this effect.
* Section 4 suggests lowering $\beta_{2}$ mitigates AdamW’s anti-regularization.   Could you provide a simple rule of thumb (e.g., $\beta_{2} \le 0.9$ when $\beta_{1} \ge 0.9$ that practitioners can adopt?

**Ethical Concerns:**

["NO or VERY MINOR ethics concerns only"]

**Final Justification:**

Given the technical novelty of the work, the constructive nature of the rebuttal, and the fact that the raised concerns have been addressed effectively, I have raised my recommendation to support the acceptance of this paper.

**Limitations:**

Yes

**Quality:**

3

**Strengths And Weaknesses:**

__Strengths__
* Presents a clear, unified framework that links optimizer memory to an explicit correction term, turning theory into a practical tool for understanding momentum and adaptive methods.
* Delivers rigorous one-step and global \mathcal{O}(h^{2}) error bounds with consistent, well-structured notation.
* Includes an intuitive worked example (Heavy-Ball) and concise tables that map popular optimizers—AdamW, NAdam, Lion—into the framework.
* Provides empirical checks on ResNet-50 (CIFAR-10) and Transformer-XL (WikiText-2) that mirror the theoretical predictions.
* Offers a principled explanation for Lion’s stronger generalization relative to AdamW and reveals its connection to Signum.

__Weaknesses__
* All proofs rely on the small-step limit h→0; the approximation gap at realistic learning rates is not quantified.
* While the theoretical analysis is strong, the experiments are restricted to small-scale benchmarks (e.g., CIFAR-10, WikiText-2). Larger or more diverse benchmarks (e.g., ImageNet, LLaMA-style pretraining) would strengthen the claims.
* Comparative advantage of memory vs. memoryless formulations is discussed but not deeply quantified—e.g., what happens when the correction term becomes unstable or insufficient? This could benefit from clearer numerical illustrations.
* The analysis focuses only on exponentially decaying memory; optimizers with long-range memory or non-exponential kernels (e.g., RMSprop or AdaBelief variants) are not covered.
* Some mathematical expressions (e.g., nested sums in Eq. 9) are dense; a schematic diagram would improve readability.
* Experimental plots lack error bars, limiting the reliability of the empirical support.

---

> ### Author Rebuttal · Authors · 2025-07-29
>
> Please accept our gratitude for your careful evaluation of our work. We answer each of your questions below.
>
> **Q1.** Thank you for your suggestion. Since we are unable to insert pictures, please refer to the list of $\\|\cdot\\|\_{\infty}$ norms of the weight differences as a function of learning rates $10^{-4}, 2 \times 10^{-4}, \ldots, 10^{-3}$, after four full epochs. The weight decay was chosen as in the paper ($10^{-3} / \text{lr}$).
> ```
> Adam, 1-order: [9.332224726676941e-05, 0.0003106476506218314, 0.0005798707134090364, 0.0007627895683981478, 0.001006735023111105, 0.0013587186112999916, 0.0015425472520291805, 0.0017450712621212006, 0.0020120255649089813, 0.0022523682564496994]
> Adam, 2-order: [3.123283386230469e-05, 0.0001106560230255127, 0.00033054023515433073, 0.0004508025012910366, 0.0007957816123962402, 0.0010535456240177155, 0.001970142126083374, 0.001563912257552147, 0.0022364631295204163, 0.0025330185890197754]
>
> Lion, 1-order: [8.964911103248596e-05, 0.0003637117915786803, 0.0005911120679229498, 0.0008356142789125443, 0.0012425384484231472, 0.0017505818977952003, 0.0017895549535751343, 0.0022158119827508926, 0.002560459077358246, 0.0030463594011962414]
> Lion, 2-order: [2.2292137145996094e-05, 7.791072130203247e-05, 0.00027696078177541494, 0.0003433115780353546, 0.000695347785949707, 0.0009671555017121136, 0.0017019212245941162, 0.0012936657294631004, 0.0019430406391620636, 0.0027693919837474823]
> ```
> These numerical results indicate that the first- and second-order approximations become the same scale at the end of the range at $h=10^{-3}$ , but we do not expect the approximation to break down within this range of learning rates.
>
>
> **Q2.** Thank you for this important question. Usually, algorithms of the AdaGrad family use exponential moving averages of past gradient matrices (arXiv:2309.06497). For example, Shampoo can be defined as
> \begin{align*}
>   \boldsymbol{L}\_t &= \big(\beta \boldsymbol{L}\_{t - 1} + (1 - \beta) \boldsymbol{G}\_t \boldsymbol{G}\_t^T\big) / (1 - \beta^{t + 1}),\\\\
>   \boldsymbol{R}\_t &= \big(\beta \boldsymbol{R}\_{t - 1} + (1 - \beta) \boldsymbol{G}\_t^T \boldsymbol{G}\_t\big) / (1 - \beta^{t + 1}),\\\\
>   \boldsymbol{W}\_{t + 1} &= \boldsymbol{W}\_t - h \cdot \boldsymbol{L}\_t^{- 1 / 4} \boldsymbol{G}\_t \boldsymbol{R}\_t^{- 1 / 4},
> \end{align*}
> where $\boldsymbol{G}\_t$ is the gradient matrix, $h$ is the learning rate. Because of the exponential moving averaging in $\boldsymbol{L}\_t$ and $\boldsymbol{R}\_t$, memory decays, and given sufficient smoothness of the loss this algorithm is indeed covered by our theory (similarly to Adam).
> We did not include such algorithms in our examples for the reasons of space, and since a complete qualitative analysis of the resulting approximation is non-trivial, likely warranting a separate article. However, we will make sure to add a discussion of this in the relevant section, following your suggestion.
>
> **Q3.** Although mini-batch Adam is automatically covered by Theorem 3.2 and Corollary 3.3, our Section 4 is devoted to full-batch comparisons because the qualitative analysis of the mini-batch noise effects on Adam and Lion in this framework requires extensive calculations. We have largely completed them and plan to present it soon as upcoming work. A summary is as follows: while in the large-batch case memory of Adam with $\beta\_2 > \beta\_1$ always anti-regularizes, this is no longer true for smaller batches, and in fact increasing $\beta\_2$ can improve generalization if the noise is sufficiently large. We predict that the threshold of the batch size at which this shift happens is of the same scale as the critical batch size (arXiv:1812.06162). We are conducting extensive experiments verifying theoretical predictions in this setting. Empirical verification of the extra regularization by noise in the simpler setting of GD (with or without momentum) can be found in earlier works [22, 51] devoted to stochastic training: they have a different focus theoretically and different proof strategies, but our interpretation is consistent with their experiments. Thank you for pointing out this issue; we will add a discussion of this.
>
> **Q4.** In Section 2 of the supplemental appendix, we provide additional evidence that taking $\beta\_2 < \beta\_1$ can strongly regularize large-batch training on vision tasks, moving the model parameters to much flatter regions of the loss space. However, overdoing this can cause divergence. As a simple rule of thumb, we would suggest taking $\beta\_1 = \beta\_2$ (e.g., $0.9$) as a first conservative option for large-batch training.
>
> Please let us know if we inadvertently missed something in your original review and/or if you have follow-up questions or suggestions that you would like us to further address. Thank you!

---

### Official Review · Reviewer_dFWS · 2025-07-03

**Clarity:** 2
**Significance:** 3
**Originality:** 3
**Rating:** 4
**Confidence:** 4

**Summary:**

This paper investigates optimization algorithms using a general update rule:

$$\theta^{(n+1)} = \theta^{(n)} - h F^{n}\left( \theta^{n}, \theta^{(n-1)}, \ldots , \theta^{0} \right) \quad (1)$$

Here, $h$ represents the step size. This formulation is broad enough to encompass numerous well-known algorithms that leverage past gradient information, including methods like Momentum, Nesterov acceleration, Adam, and Lion. The paper approximates (1) with a memoryless algorithm operating on a modified loss function. This modification reveals how memory regularizes the optimization process. The techniques developed are then applied to study the implicit regularization properties of Adam and Lion. This analysis shows that, with certain hyperparameter configurations, Adam can actually anti-regularize the optimization dynamics.

**Questions:**

1) Can the authors comment on Assumption 3.1 ? How restricitve is to assume that the coefficient decay, is it possible to verify this for example for Adam (or even momentum GD) in some simpler settings ?

**Ethical Concerns:**

["NO or VERY MINOR ethics concerns only"]

**Final Justification:**

The rebuttal from the authors gave me confidence that the paper will be revised to improve technical clarity and also more detailed experimental section. However, given some restrictive assumptions, I maintain my initial positive rating and do not raise it further.

**Limitations:**

yes

**Paper Formatting Concerns:**

No major formatting concerns.

**Quality:**

3

**Strengths And Weaknesses:**

Strengths:

a) One of the strengths of the paper is to provide an unified framework to study the regularization properties of various optimization algorithms that utilize the memory. This makes the techniques applicable to many scenarios.

b) The intuition and the derivation of the result is well incorporated in the main text with an illustrating example of momentum GD.

c) Identifying that the small $\beta_2$ of Adam leads to better performance is a interesting result and can give some insights into how to  choose the hyperparameters.

Weakness:

a) A significant weakness of the results lies in the lack of precise statements regarding the implicit regularization of Adam and Lion within Section 5. For instance, lines 176 and 191 contain phrases such as "Neglecting coefficients decaying to zero exponentially fast." Such statements require formalization to clearly define the regime of applicability for the results. This precision would also facilitate a more direct comparison with prior work, such as [56], which investigates other implicit regularization aspects of Adam. It is highly recommended to include formal Lemmas for the implicit regularization of Adam and Lion, detailing all necessary technical simplifications.

b) The empirical evaluation section presents several drawbacks:
- Limited Scope of Trajectory Comparison: The comparison of trajectories is exclusively restricted to MLPs. Extending this analysis to other network architectures would provide a more comprehensive understanding. Additionally, the inclusion of error bars for these comparisons would be beneficial for assessing the robustness of the observed trends.
- Learning Rate in Figure 2: Figure 2 should be further investigated with varying learning rates. This would help determine if the observations concerning $\beta_1$ and $\beta_2$ generalize across different learning rate schedules.

---

> ### Author Rebuttal · Authors · 2025-07-29
>
> Thank you very much for your thoughtful review and suggestions.
>
> Regarding your question and weakness (a), we have added the two lemmas given below (and their proofs) to the next revision. In particular, classical Adam is indeed covered by our framework. The assumption on the decay of memory appears to be mild since usually memory decays exponentially.
>
> The lemma for Adam is as follows.
>
> *Lemma 1.* Let $\{\boldsymbol{\theta}^{(n)}\}\_{n \in \mathbb{Z}\_{\geq 0}}$ be the sequence of vectors generated by the iteration in Equation (1) with initial condition $\boldsymbol{\theta}^{(0)}$, where $\boldsymbol{F}^{(n)}(\cdot)$ is as defined in Example 1.3, and the loss function $\mathcal{L}(\cdot)$ defined in an open convex bounded domain $\mathcal{D} \subset \mathbb{R}^d$ is three times continuously differentiable with bounded derivatives; also, let $T \geq 0$ be a fixed ``time'' horizon. Then Assumption 3.1 holds; in particular, by Corollary 3.3 the inequality $\max\_{n \in [0:\lfloor T / h \rfloor]} \bigl\\| \boldsymbol{\theta}^{(n)} - \tilde{\boldsymbol{\theta}}^{(n)} \bigr\\|\_\infty \leq C\_2 h^2$ holds, where
> \begin{align*}
>   &\tilde{\theta}\_r^{(n + 1)} = (1 - \lambda h) \tilde{\theta}\_r^{(n)} - h \bigg(\frac{\partial\_r \mathcal{L}\big(\tilde{\boldsymbol{\theta}}^{(n)}\big)}{\big(\big|\partial\_r \mathcal{L}\big(\tilde{\boldsymbol{\theta}}^{(n)}\big)\big|^2 + \varepsilon\big)^{1 / 2}} + M\_{r}^{({n})}\big(\tilde{\boldsymbol{\theta}}^{(n)}\big)\bigg),\\\\
>   &M\_{r}^{({n})}\big(\boldsymbol{\theta}\big) =
>     h \Bigg(
>                            \frac{\beta\_1}{1 - \beta\_1} - \frac{(n + 1) \beta\_1^{n + 1}}{1 - \beta\_1^{n + 1}}
>                            -
>                            \frac{\beta\_2}{1 - \beta\_2} + \frac{(n + 1) \beta\_2^{n + 1}}{1 - \beta\_2^{n + 1}}
>                            + \varepsilon \frac{\frac{\beta\_2}{1 - \beta\_2} - \frac{(n + 1) \beta\_2^{n + 1}}{1 - \beta\_2^{n + 1}}}{|\partial\_r \mathcal{L}(\boldsymbol{\theta})|^2 + \varepsilon}
>                            \Bigg) \frac{\big(\partial\_r \\| \nabla \mathcal{L}(\boldsymbol{\theta}) \\|\_{1, \varepsilon} + \lambda [\nabla^2 \mathcal{L}(\boldsymbol{\theta}) \boldsymbol{\theta}]\_r\big)}{(|\partial\_r \mathcal{L}(\boldsymbol{\theta})|^2 + \varepsilon)^{1 / 2}},\quad r \in [1:d].
> \end{align*}
>
> The lemma for Lion is as follows.
>
> *Lemma 2.* Let $\{\boldsymbol{\theta}^{(n)}\}\_{n \in \mathbb{Z}\_{\geq 0}}$ be the sequence of vectors generated by the iteration in Equation (1) with initial condition $\boldsymbol{\theta}^{(0)}$, where $\boldsymbol{F}^{(n)}(\cdot)$ is as defined in Example 1.5, the function $\mathcal{K}(\cdot): \mathcal{M} \to \mathbb{R}$ defined in a open bounded region $\mathcal{M} \subset \mathbb{R}^d$ is three times continuously differentiable with bounded derivatives, and the loss function $\mathcal{L}(\cdot)$ defined in an open convex bounded domain $\mathcal{D} \subset \mathbb{R}^d$ is three times continuously differentiable with bounded derivatives; also, let $T \geq 0$ be a fixed ``time'' horizon. Then Assumption 3.1 holds; in particular, by Corollary 3.3 the inequality $\max\_{n \in [0 : \lfloor T / h \rfloor}] \big\\|\boldsymbol{\theta}^{(n)} - \tilde{\boldsymbol{\theta}}^{(n)}\big\\|\_\infty \leq C\_2 h^2$ holds, where
> \begin{align*}
>   &\tilde{\theta}\_r^{(n + 1)} = (1 - \lambda h) \tilde{\theta}\_r^{(n)} - h \big(- \partial\_r \mathcal{K} \big(- (1 - \rho\_1 \rho\_2^n) \nabla \mathcal{L}(\tilde{\boldsymbol{\theta}}^{(n)})\big) + M\_{r}^{({n})}\big(\tilde{\boldsymbol{\theta}}^{(n)}\big)\big),\\\\
>   &M\_{r}^{({n})}\big(\boldsymbol{\theta}\big) = h \rho\_1 (1 - \rho\_2) \sum\_{i = 1}^d \sum\_{j = 1}^d \partial\_{j r} \mathcal{K} \big(- (1 - \rho\_1 \rho\_2^n) \nabla \mathcal{L}(\boldsymbol{\theta})\big) \partial\_{i j} \mathcal{L}(\boldsymbol{\theta}) \sum\_{k = 1}^n \rho\_2^{k - 1} \sum\_{s = n - k}^{n - 1} \big(- \partial\_i \mathcal{K} \big(- (1 - \rho\_1 \rho\_2^s) \nabla \mathcal{L}(\boldsymbol{\theta})\big) + \lambda \theta\_i\big)\\\\
>   &\quad = h \frac{\rho\_1}{1 - \rho\_2} \sum\_{i = 1}^d \sum\_{j = 1}^d \partial\_{j r} \mathcal{K} \big(- \nabla \mathcal{L}(\boldsymbol{\theta})\big) \partial\_{i j} \mathcal{L}(\boldsymbol{\theta}) \big(- \partial\_i \mathcal{K} \big(- \nabla \mathcal{L}(\boldsymbol{\theta})\big) + \lambda \theta\_i\big) + o\_n(1),\quad r \in [1:d].
> \end{align*}
> where $o\_n(1)$ is a function of $\boldsymbol{\theta}$ converging to zero uniformly in $\boldsymbol{\theta}\in \mathcal{D}$.
>
> ----------
>
> Regarding weakness (b), we are conducting sweeps with more learning rates; in addition, we would like to refer to Section 2 ``Additional Experiments'' in the supplementary material (file `supplemental_appendix`) where two more learning rates for each of the tasks (vision and language) are provided; we will expand the sweeps and present them more noticeably in the paper. For the trajectory comparison in the weight space, we have run additional simulations with a small CNN and a simple Vision Transformer, following your suggestion. In the CNN case, the corresponding weight difference $\infty$-norms for Adam with $h = 10^{-3}$ after $1, \ldots, 5$ full epochs are as follows:
> ```
> 1-order: [2.9802322387695312e-08, 2.261623740196228e-05, 6.423518061637878e-05, 0.00012150406837463379, 0.0001912824809551239]
> 2-order: [2.9802322387695312e-08, 1.644715666770935e-06, 7.167458534240723e-06, 1.864507794380188e-05, 3.791600465774536e-05]
> ```
>
> and $h = 3 \times 10^{-4}$:
> ```
> 1-order: [2.9802322387695312e-08, 2.08243727684021e-06, 6.068497896194458e-06, 1.1827796697616577e-05, 1.9203871488571167e-05]
> 2-order: [2.9802322387695312e-08, 4.470348358154297e-08, 1.4156103134155273e-07, 3.9674341678619385e-07, 8.754432201385498e-07]
> ```
>
> For the ViT, Adam with $h = 10^{-4}$:
> ```
> 1-order: [5.960464477539063e-08, 9.341537952423096e-05, 0.00017436407506465912, 0.0003153085708618164, 0.00037641823291778564]
> 2-order: [5.960464477539063e-08, 3.9421021938323975e-05, 0.00014676153659820557, 0.0004404708743095398, 0.000798331166151911]
> ```
> and $h = 3 \times 10^{-5}$:
> ```
> 1-order: [5.960464477539063e-08, 1.2260396033525467e-05, 3.2086391001939774e-05, 4.717335104942322e-05, 6.285309791564941e-05]
> 2-order: [5.960464477539063e-08, 2.4118926376104355e-06, 8.469447493553162e-06, 2.0730774849653244e-05, 3.071478568017483e-05]
> ```
> We will also add error bars to Figure 1 as you helpfully suggested.
>
> Please let us know if you have additional comments or follow-up questions which we will be happy to address. Thank you!

---

### Official Review · Reviewer_WQ9W · 2025-07-03

**Clarity:** 3
**Significance:** 3
**Originality:** 3
**Rating:** 5
**Confidence:** 4

**Summary:**

The paper contains a theory to show how gradient-based algorithms with momentum
terms, and more generally optimization algorithms that have a non-local in time
update rule but whose dependence on past-in-time information decays exponentially,
can be approximated by a local-in-time update rule. The paper gives a general
closed-form derivation of this memoryless update rule.

**Questions:**

1. Equation (6), if I interpret correctly, is obtained in the limit
$n\to\infty$.  But how can we be sure that the initial iterations do not
substantially change the trajectory in the parameter space, giving us
substantially different methods?

2. It is well known that gradient descent with momentum (and all the other
modern gradient-based variations like Adagrad, Adam, RMSprop etc) can be
interpreted in terms of a sequence of optimization problems; this is the
typical view that theories like Minimizing Movements, proximal methods and
mirror descent offer. See for instance section 3 of this work

Betti, A., Ciravegna, G., Gori, M., Melacci, S., Mottin, K., & Precioso,
F. (2024). A new perspective on optimizers: leveraging Moreau-Yosida
approximation in gradient-based learning. Intelligenza Artificiale, 18(2),
301-311

Do you think that there could be direct connections between these two
methods, for instance between the modified "effective" loss that you compute
and the regularized loss that is used to define GD with momentum in this
framework?

3. Can you comment on where the condition in Eq (7) is used? Is it used in
the direct derivation of the method or in the approximation bound result?

4. It is not clear to me how this analysis and the error bound could carry
over to the stochastic counterpart of the described optimization algorithms
that are fundamental in ML.

5. In the derivation of GD with momentum in section 2 and in the general
case, it is not completely evident why in the equation after line 127 the
term $\sum_{k=0}^n O(k^2 h^2)$ could not give contributions of order $O(1)$
for big values of $n$ and $k$. How can we be sure that we can discard these
terms?

**Ethical Concerns:**

["NO or VERY MINOR ethics concerns only"]

**Final Justification:**

I've raised my rating as a result of the answers provided by the authors. I have carefully read the authors’ rebuttal as well as the comments from the other reviewers and the AC. I did not find any additional major issues raised by the other reviewers. For these reasons, I recommend acceptance of this work.

**Limitations:**

yes

**Quality:**

3

**Strengths And Weaknesses:**

The article is extremely well written and clear, and the contribution, to the
best of my knowledge, is novel. Even though it does not present a new model
or optimization scheme, it provides interesting insights into interpreting
well-known and widely used optimization algorithms.

In particular, it seems to offer a useful theoretical tool to better
understand and highlight different behaviors of a class of optimization
algorithms. On the other hand, if I understand correctly, it does not provide
an alternative, implementable, viable relaxation or approximation, since the
modified loss or the correction term typically contains higher-order
derivatives that are hard to compute numerically (Eq. (6) and Eq. (9)).

The analysis of AdamW and Lion in Section 4 is interesting.

The comments in Section~6.1 are particularly appealing to me, as they seem to
indicate (and please correct me if I have misunderstood) that, for instance,
the Heavy Ball method can be approximated (in the continuous limit) by a
gradient flow in which the "energy function" already contains the gradient of
the original loss.

I liked the paper and am willing to further raise my rating after the
rebuttal period, based on the answers to my comments and questions.

---

> ### Author Rebuttal · Authors · 2025-07-29
>
> Thank you for your careful reading of our paper and valuable comments. We answer each of your questions below.
>
> **Q1.** It is a very good point which we thank you for making. Technically, it is impossible to obtain an $O(h^2)$ global approximation of GD with momentum by an ordinary GD independent of $n$. This is why our iteration given in Equation (4) depends on $n$. However, qualitatively, the dynamics becomes close to a fixed ordinary GD, and the goal of neglecting $o_n(1)$ coefficients was only to simplify the description of this qualitative behavior, as common in the literature (in particular, [22] discusses heavy-ball momentum). This is related to the fact that momentum methods can be approximated by gradient flow on an exponentially attractive invariant manifold but not everywhere, as discussed in Section 4 of [31]. As you point out, the trajectories are not the same, and we will add into the Appendix precise lemmas with the $o_n(1)$ terms given explicitly. The lemma pertaining to heavy-ball momentum is as follows:
>
> *Lemma.* Let $(\boldsymbol{\theta}^{(n)})\_{n \in \mathbb{Z}\_{\geq 0}}$ be the sequence of vectors generated by the iteration in Equation (1) with initial condition $\boldsymbol{\theta}^{(0)}$, where $\boldsymbol{F}^{(n)}(\cdot)$ is as defined in Example 1.1, and the loss function $\mathcal{L}(\cdot)$ defined in an open convex bounded domain $\mathcal{D} \subset \mathbb{R}^d$ is three times continuously differentiable with bounded derivatives; also, let $T \geq 0$ be a fixed "time" horizon. Then Assumption 3.1 holds; in particular, by Corollary 3.3 the inequality $\max\_{n \in [0:\lfloor T / h \rfloor]} \big\\|\boldsymbol{\theta}^{(n)} - \tilde{\boldsymbol{\theta}}^{(n)}\big\\|\_\infty \leq C_2 h^2$ holds, where
> $$
>   \tilde{\theta}\_r^{(n + 1)} =  \tilde{\theta}\_r^{(n)} - h \bigg(
>     \frac{1 - \beta^{n + 1}}{1 - \beta} \partial\_r \mathcal{L}\big(\tilde{\boldsymbol{\theta}}^{(n)}\big)
>     + M\_{r}^{({n})}\big(\tilde{\boldsymbol{\theta}}^{(n)}\big)\bigg),
> $$
> $$
>   M_{r}^{({n})}\big(\boldsymbol{\theta}\big) =  h \frac{\beta [1 - (2 n + 1) \beta^n (1 - \beta) - \beta^{2 n + 1}]}{2 (1 - \beta)^3} \partial_r \\|\nabla \mathcal{L}(\boldsymbol{\theta})\\|^2,\quad r \in [1:d].
> $$
>
> **Q2.** While it is not currently clear how to relate the modified loss with the minimization problem in Eq. (22) in the paper you provided, there are some connections to be noticed. For example, GD with momentum can be seen as a smoothed-out version of GD which is steepest descent with respect to the 2-norm (as written in Eq. (8) in that paper). Our framework informs that memory implicitly penalizes the squared two-norm of the gradient $\\| \nabla \mathcal{L}(\boldsymbol{\theta}) \\|^2$ which is the square of $\min\_{\|\boldsymbol{\\delta}\\| \leq 1} \nabla \mathcal{L}(\boldsymbol{\theta})^T \boldsymbol{\delta} \approx \min_{\\|\boldsymbol{\delta}\\| \leq 1} \mathcal{L}(\boldsymbol{\theta} + \boldsymbol{\delta})$; the latter is the same optimization problem as the one below Eq. (3) at the end of the column in that paper. The same argument can be made about Adam (smoothed-out version of sign descent, steepest descent under the infinity norm) and a different pair of dual norms $\\| \cdot \\|_{\infty}$ and $\\| \cdot \\|_1$. We currently do not have a complete understanding of these coincidences but we plan to add a discussion of this as a future direction and we thank you for pointing it out.
>
> **Q3 and Q5.** Equation (7) is used to control the accumulation of error: intuitively, since memory decays fast enough, past errors also decay fast enough. For the special case of GD with momentum, consider the term $O(k^2 h^2)$ after line 127 you referred to. The $k^2$ factor in the error disappears because $\sum_{k = 0}^n \beta^k O(k^2 h^2) = O(h^2)$ by virtue of summability $\sum_{k = 0}^\infty \beta^k k^2 < \infty$. Equation (7) generalizes this and is used in similar contexts, such as before line 151, in lines 155-156 in our derivation section, in line 663, between lines 668-669 in the proof.
>
> **Q4.** You are correct in pointing out that stochasticity introduces additional challenges. Theorem 3.2 and Corollary 3.3 are applicable regardless of whether the loss function is the same at each iteration or different as in mini-batch training, so the technical approximation result can be obtained for mini-batch training with no additional work. However, one then needs to choose a framework to analyze the results qualitatively, e.g. identify implicit regularization by noise. We illustrate that is possible in Section 6.2 for the simple special case of stochastic GD with momentum. While not in the scope of this paper, we know that it is possible for AdamW as well (although with significantly longer calculations), and we plan to present it as upcoming work.
>
> Please let us know if we accidentally misunderstood or missed any of your points or you have additional questions or suggestions. Thank you!

---

> > ### Comment · Reviewer_WQ9W · 2025-08-05
> >
> > Thank you for the very clear answers to my questions. It’s refreshing to see a paper that follows what used to be the hallmark of scientific writing: introducing a meaningful idea, analyzing it theoretically, and validating it experimentally — something that, unfortunately, is becoming less common.
> >
> > I’m particularly intrigued by the potential connections with Minimizing Movements, and I look forward to reading more about your perspective on this. Hopefully, after the review process is complete, there will be an opportunity to discuss it further.
> >
> > As promised, I will raise my rating.

---

> > > ### Author Response · Authors · 2025-08-07
> > >
> > > Thank you very much for your encouraging comments about our work! We will certainly look into the potential connections with Minimizing Movements in future research.

---

### Note · Authors · 2025-08-12

We extend our gratitude to all reviewers for their valuable feedback. We are encouraged by the positive comments the reviewers gave about the paper, including the remarks about the generality of our framework (Reviewers dFWS, Fq9j), rigor and well-structured notation (Reviewer Fq9j), and clarity (Reviewers WQ9W, Fq9j, CQS8). We hope we have addressed the points raised by the reviewers as a result of the discussion. In particular, following suggestions from Reviewers WQ9W and dFWS, we have included into the next revision additional lemmas making correction terms more precise (for finite $n$) in our illustrative examples; following suggestions from Reviewers dFWS, Fq9j, CQS8 we have included additional empirical results. We also thank all reviewers for their insightful questions which lead to future research directions building on our proposed framework, such as the question about Minimizing Movements by Reviewer WQ9W, about AdaGrad and Shampoo by Reviewer Fq9j, and about combining implicit biases by Reviewer CQS8. We are also grateful to the Area Chair for the consideration of our paper.

---

### Decision · Program_Chairs · 2025-09-17

**Decision:**

Accept (poster)

**Comment:**

This paper studies the impact of the dependence on past iterates (such as momentum), i.e. memory, in optimization algorithms.
In particular, it is shown that an optimization with memory can be approximated by a memory-less one by introducing an additional correction term, which can be interpreted as a perturbation of the loss.
The authors apply the proposed framework to analyze various optimization algorithms, including AdamW, Lion and Signum.

The paper is well-written and provides insights for understanding the role of memory in optimization algorithms, receiving unanimous support from the reviewers.